# RHOA signaling defects result in impaired axon guidance in iPSC-derived neurons from patients with tuberous sclerosis complex

Timothy S. Catlett[1], Massimo M. Onesto[1], Alec J. McCann[1], Sarah K. Rempel[1], Jennifer Glass[2], David N. Franz[2] & Timothy M. Gómez ⬡ [1✉]

Patients with Tuberous Sclerosis Complex (TSC) show aberrant wiring of neuronal connections formed during development which may contribute to symptoms of TSC, such as intellectual disabilities, autism, and epilepsy. Yet models examining the molecular basis for axonal guidance defects in developing human neurons have not been developed. Here, we generate human induced pluripotent stem cell (hiPSC) lines from a patient with TSC and genetically engineer counterparts and isogenic controls. By differentiating hiPSCs, we show that control neurons respond to canonical guidance cues as predicted. Conversely, neurons with heterozygous loss of TSC2 exhibit reduced responses to several repulsive cues and defective axon guidance. While TSC2 is a known key negative regulator of MTOR-dependent protein synthesis, we find that TSC2 signaled through MTOR-independent RHOA in growth cones. Our results suggest that neural network connectivity defects in patients with TSC may result from defects in RHOA-mediated regulation of cytoskeletal dynamics during neuronal development.

[1] Department of Neuroscience, University of Wisconsin School of Medicine and Public Health, Madison, WI, USA. [2] Department of Pediatrics, Cincinnati Children's Hospital Medical Center, Cincinnati, OH, USA. ✉email: tmgomez@wisc.edu

Accurate guidance of developing axons is a critical early process necessary for the correct assembly of neural circuits. The tips of these axonal projections, growth cones, follow patterns of chemical and mechanical cues to extend along pathways to reach specific synaptic partners. Each growth cone expresses receptors to these various cues and integrates multiple discrete signals to ultimately effect axon extension and guidance through cytoskeletal rearrangements[1–3]. This guidance cue-to-cytoskeleton signaling axis involves numerous intermediaries, which have been shown in animal model neurons to include both the Rho family GTPases and local protein synthesis (LPS) within the growth cone[4,5]. While defects in signal transduction have been linked to abnormal neural network wiring in animal models, few human neural development disorders have yet been attributed to specific growth cone signaling abnormalities[6–8].

Tuberous sclerosis complex (TSC) is an autosomal dominant neurodevelopmental disorder affecting approximately one million people worldwide. Most TSC patients have some degree of neurological symptoms including mild to profound intellectual disability (50%), autism spectrum disorders (ASDs) (50%), and up to 90% develop seizures[9]. Pathological variants in the TSC1 (hamartin) or TSC2 (tuberin) genes cause TSC, which is best characterized by benign tumors called hamartomas that form within multiple organ systems, including the central nervous system (CNS)[10,11]. While neurological symptoms were classically attributed to the formation of cortical tubers during embryonic development, recent research indicates there is a poor correlation between CNS tubers and the severity of epilepsy and intellectual disability. While the formation of tumors in TSC has long been attributed to the contribution of a "second-hit" loss of the normal TSC1/2 allele, recent research has shown that many tumors, especially cortical tubers, lack a second-hit, raising the possibility of an alternate pathway for tumorigenesis that depends on monoallelic inactivation of TSC1/2[12]. In addition, TSC patients with severe intellectual disabilities may have defects in neural network wiring due to abnormal axon guidance and synaptogenesis during development. Consistent with this notion, diffusion tensor imaging (DTI) studies show abnormal axon tract development in patients with mutations in TSC1 or TSC2[13–15]. Neural connectivity abnormalities are also likely responsible for some learning and behavioral abnormalities in TSC animal models, which typically lack tubers[16,17]. Moreover, one group has shown both retinotectal axon guidance abnormalities and guidance cue insensitivity in a heterozygous Tsc2 mutant mouse model[18].

Here, we demonstrate that developing human neurons derived from unaffected and disease-corrected iPSCs are sensitive to and can be guided by several canonical axon guidance cues, including ephrin-A1, ephrin-A5, slit-2, netrin-1, and semaphorin-3F. In contrast, TSC patient-derived neurons from TSC2[+/−] mutant iPSCs exhibit enhanced basal axon extension and severe insensitivity to several guidance cues. Interestingly, basal MTOR activity and MTOR-mediated protein synthesis modulation in response to cues is normal in TSC2[+/−] mutant neurons, suggesting that defective MTOR signaling may not account for abnormal axon growth and guidance phenotypes. In contrast, basal and cue-activated RHOA was diminished in TSC2[+/−] neurons, suggesting that TSC1/2 signaling through RHOA is necessary for proper axon extension and response to guidance cues in these neurons. Consistent with this notion, we find that mTORC1 and mTORC2 inhibitors neither significantly impact control neuron outgrowth nor rescue axon growth and guidance defects in TSC2[+/−] neurons. Therefore, contrary to reports in some animal model systems, we find that human neurons do not require rapid TSC2-dependent modulation of protein synthesis to regulate outgrowth and cue responses. Instead, neurons require direct modulation of RHOA signaling downstream of TSC1/TSC2 for proper axon development.

## Results

### Generation and gene editing of iPSC lines from a TSC patient.
Fibroblasts were isolated from skin biopsies from an 18-year-old male TSC patient at the Tuberous Sclerosis Clinic at Cincinnati Children's Hospital Medical Center. This patient has a history of complex partial seizures, autism and expressive language delay, intellectual disability, as well as a large tuber burden. Genomic sequencing confirmed a heterozygous C>G point mutation at bp 972, which results in a premature stop at codon 324 (of 1843). Patient-derived fibroblasts were reprogrammed into iPSCs using non-integrating Sendai virus-containing genes for the transcription factors OCT4, SOX2, KLF4, and MYC (Invitrogen Cyto-Tune™-iPS 2.0 Sendai Reprogramming Kit.) by the Waisman Center iPSC service core (UW-Madison). This service includes verification of stem cell marker expression, chromosome karyotyping, and Short Tandem Repeat (STR) analysis to confirm that the somatic cells and derived iPSCs are from the same individual and each line is a unique clone without contamination. Subsequently, we used CRISPR-Cas9 genomic editing to correct the point mutation in TSC2[+/−] iPSCs to generate isogenic control TSC2[+/+] iPSCs and also introduced the same mutation in the unaffected allele to create null TSC2[−/−] iPSCs via CRISPR-Cas9. We also used CRISPR-Cas9 to generate the same heterozygous C>G point mutation at bp 972 in a second TSC2[+/−] iPSC line within IMR90 control iPSCs. Two clones of each respective patient genotype, in addition to a clone from the IMR90[TSC2+/−] cell line, were used in experiments; stem cell colonies exhibited normal morphologies and karyotypes, divided at similar rates, and exhibited pluripotency markers (Supplemental Fig. 1a, b).

### TSC2 haploinsufficiency does not alter basal MTOR activity in cortical neurons.
Excitatory glutamatergic human forebrain neurons (hFB) of each TSC2 genotype were differentiated using established protocols[19,20] and plated as neurospheres for 24 h before fixation and immunolabeling. Developing neurospheres from each independent cell line were positive for the neuroectoderm marker PAX6, the neural stem cell marker SOX2, and dorsal forebrain marker FOXG1. (Supplementary Fig. 1C, D). Axonal projections were identified due to their extreme lengths relative to dendritic projections, and growth cones less than 100 microns from the edge of the neurosphere were excluded from all analysis. To confirm axonal identity, we immunostained neurons at 1 day in vitro (DIV) with the axonal marker TAU, which revealed strong labeling of all long projections and very weak labeling by the dendritic marker MAP2. At 30DIV, strong MAP2 staining was observed in short dendritic projections in addition to the long TAU-labeled axonal projections (Supplementary Fig. 1E). Next, we used TSC2-specific antibodies to assess TSC2 expression within the growth cones of each genotype. We found that control neurons differentiated from corrected TSC2[+/+] iPSCs, as well as IMR90 growth cones (Supplementary Fig. 1F), exhibited intense staining for TSC2 within growth cones and throughout neurites, while TSC2[+/−] growth cones showed approximately 50% less fluorescence signal and TSC2[−/−] growth cones were devoid of labeling (Fig. 1A, B, Supplementary Fig. 1F). Similarly, neurosphere lysates immunoblotted for TSC2 showed a reduction and absence of protein in heterozygous and null hFB neurons, respectively, compared to control (Fig. 1C). To determine the effects of TSC2 loss on MTOR activity, we immunolabeled growth cones for ribosomal protein phospho-S6 (Ser235/236), a well-known marker for MTOR activation and protein synthesis[21,22]. Interestingly, while TSC2[−/−] growth cones showed

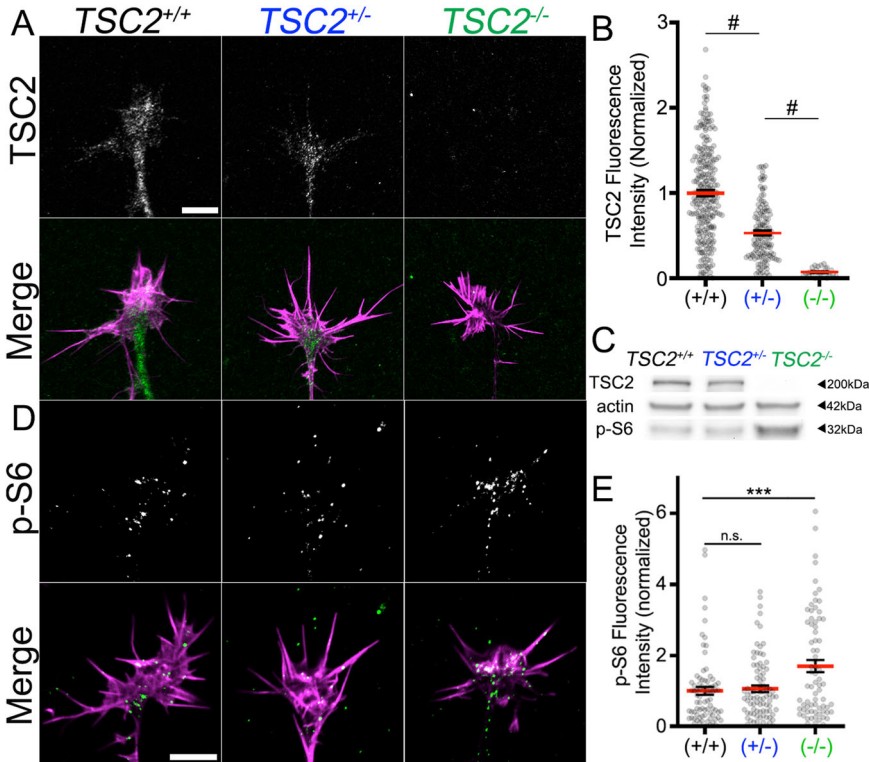

**Fig. 1 Growth cones of hFB neurons differentiated and *TSC2* patient iPSCs have reduced TSC2 expression and increased MTOR signaling. A** hFB neuronal growth cones of indicated genotype immunolabeled for TSC2 (green) and counter-stained for F-actin (magenta) in the merge. Note the complete absence of TSC2 in the null growth cone. **B** Average fluorescence intensity values of TSC2 in all measured growth cones. **C** Western blots from extracted hFB neurospheres of each genotype showed similar reduction and absence of TSC2 in heterozygous and null *TSC2* lines, respectively. Western blots also showed increased phospho-S6 (p-S6), a downstream target of mTORC1, only in *TSC2*$^{-/-}$ neurons, while total S6 was unchanged (repeated with similar results across three independent differentiations). **D** hFB neuronal growth cones of indicated genotype immunolabeled for p-S6 (green) and counter-stained for F-actin (magenta) in the merge. **E** Fluorescence intensity values of phospho-S6 immunofluorescence in all measured growth cones. Note that similar to Western blot results, P-S6 was elevated only in *TSC2*$^{-/-}$ growth cones (*TSC2*$^{+/+}$ vs. *TSC2*$^{+/-}$, $P = 0.94$, *TSC2*$^{+/+}$ vs. *TSC2*$^{-/-}$, $P = 0.0004$). ***$P < 0.001$, #$P < 0.0001$, One-way ANOVA with Tukey's Multiple Comparison, represented as mean +/− SEM. Scale, 5 μm. Additional data on all experimental groups in Supplementary Information.

significant upregulation of p-S6 levels, *TSC2*$^{+/-}$ growth cones appeared similar to *TSC2*$^{+/+}$ growth cones (Fig. 1D, E, Supplementary Fig. 1G). These immunofluorescence findings were corroborated by immunoblotting (Fig. 1C). Therefore, heterozygous loss of TSC2 does not appear to impact basal MTOR activity within growth cones nor within neurospheres under these culture conditions.

**Guidance of cortical axons by patterned inhibitory cues is defective in *TSC2*$^{+/-}$ neurons.** While stem cell-derived human neurons have been widely used for over two decades, few studies have tested whether developing human neurons are sensitive to classic axon guidance cues[23–25] and none have determined whether neurons carrying mutations in ASD-related genes respond similarly to unaffected neurons. Given that neural network connectivity abnormalities may be responsible for some TSC pathologies[13–15], we hypothesized that *TSC2*$^{+/-}$ growth cones may be less sensitive to canonical guidance cues. Here we focused on heterozygous hFB neurons because the vast majority of TSC patient neurons are heterozygous compared to a minority of cells in some brain tubers that may carry a second-hit mutation[12]. To begin to test this notion, we first needed to determine whether normal-developing cortical neurons could be guided through growth cone interactions with discontinuous patterns of inhibitory cues. hFB neurospheres were plated onto alternating stripes of laminin (LN) (25 μg/mL) and pre-clustered

ephrin-A5-Fc (10 μg/mL) or LN (control) (see "Methods" section). Ephrin-As are well-known inhibitory axon guidance cues that have important roles in neural network assembly throughout the developing nervous system[26]. After 3DIV, neurons were fixed and immunolabeled for Fc to visualize ephrin and axons (cross-reactive), as well as phalloidin to label F-actin. As seen in Fig. 2A, control axons are guided along with stripes of pure LN and are robustly repelled from stripes of ephrin-A5. On the other hand, *TSC2*$^{+/-}$ hFB neurons have long axons that crossed ephrin-A5 stripes repeatedly. We measured guidance along stripes using a modified Sholl analysis[27] by dividing each neurosphere into four quadrants (Fig. 2B, inset), then pooling intersecting axons as they traveled parallel or perpendicular to the stripes. From Sholl analysis, it is clear that *TSC2*$^{+/+}$ control neurons extend further parallel to stripes compared to perpendicular (Fig. 2B), while *TSC2*$^{+/-}$ neurons extend long axons across ephrin-A5 stripes indiscriminately (Fig. 2A) and show little preference for perpendicular vs. parallel orientation (Fig. 2B).

The total area of axons extending upon LN vs. ephrin stripes was also measured to assess substratum preference (Fig. 2C, D). Corrected control and IMR90 hFB neurons showed robust avoidance of both ephrin-A5 and ephrin-A1 stripes, while *TSC2*$^{+/-}$ patient-derived and IMR90$^{TSC2+/-}$ neurons were not significantly repelled from stripes of either ephrin (Fig. 2C, D, Supplemental Fig. 2A, B, E–H). To confirm that ephrin was responsible for avoidance behaviors, control neurons were plated upon stripes of unclustered ephrin-A5, since pre-clustering of ephrin-A5, but not ephrin-A1, is

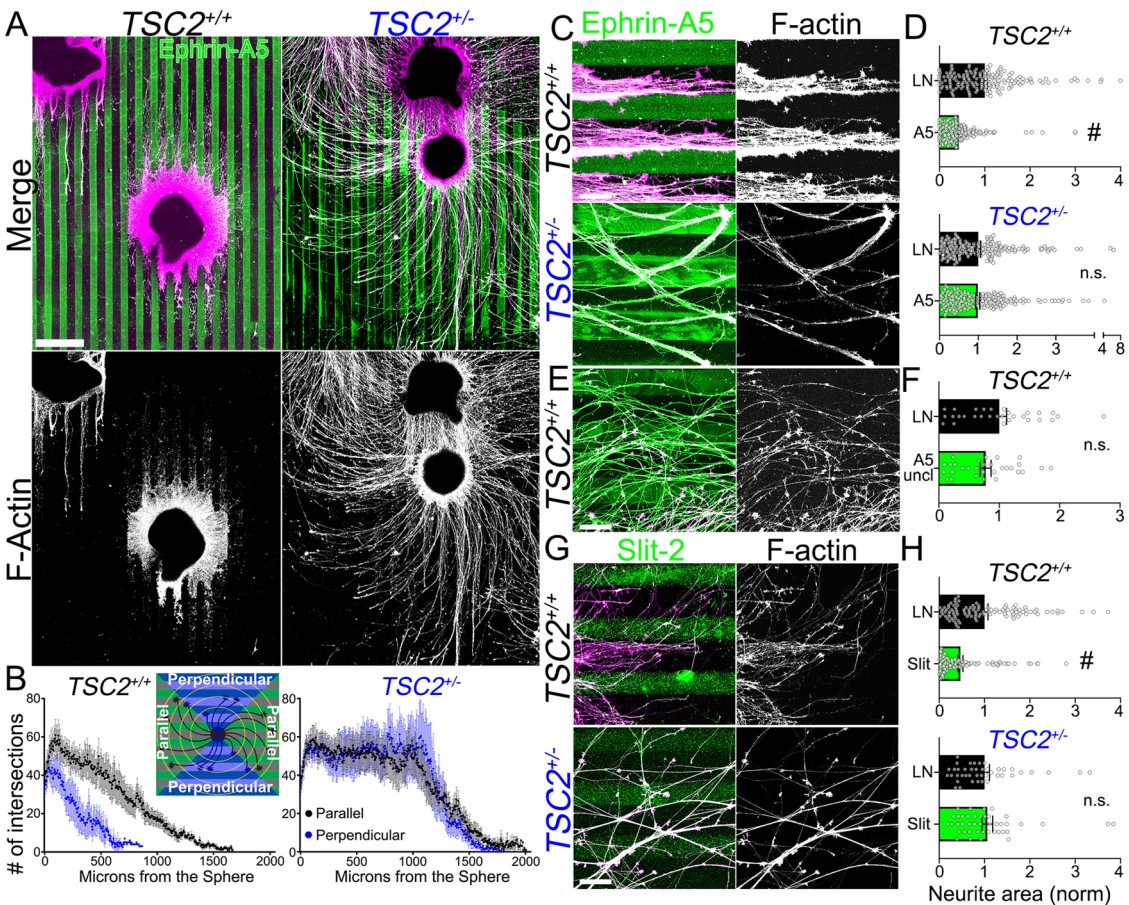

**Fig. 2 Pathfinding by control cortical neurons along ephrin-A5 and Slit-2 repulsive stripes is abnormal in *TSC2*⁺/⁻ neurons. A** hFB neurospheres were cultured for three days on parallel stripes of Fc-tagged ephrin-A5 and laminin (LN), then fixed and stained for anti-Fc (green in merges) and F-actin (magenta in merges). Note that anti-Fc antibody cross-reacts with neurites, making processes appear white in the merge. Many *TSC2*⁺/⁺ neurites extended upon LN, parallel to the pattern while avoiding the ephrin-A5 stripes. On the other hand, *TSC2*⁺/⁻ hFB neurites showed little substratum preference, crossing ephrin-A5 containing lanes repeatedly (repeated in 8 (*TSC2*⁺/⁺) and 10 (*TSC2*⁺/⁻) neurospheres across 4 differentiations; see Supplementary Information for additional data). **B** Modified Sholl analysis measures neurite crossings of concentric circles every 10 μm away from sphere edge, with data binned by quadrant for neurites extending parallel and perpendicular to stripes (inset). Graphs show mean +/− SEM of 8 and 10 independent *TSC2*⁺/⁺ and *TSC2*⁺/⁻ neurospheres, respectively. Note long *TSC2*⁺/⁻ neurites extended equally well in all directions from spheres. **C, D** High magnification images of LN/ephrin-A5 patterns (**C**) and analysis (**D**) of the thresholded area occupied by neurites (F-actin channel) on each substratum normalized to the area of neurites growing on laminin only (*TSC2*⁺/⁺ ephrin-A5 vs. laminin, *P* < 0.0001, *TSC2*⁺/⁻ ephrin-A5 vs. laminin, *P* = 0.85) (see "Methods" section). **E, F** *TSC2*⁺/⁺ hFB neurites grown on unclustered ephrin-A5 controls were not guided (*P* = 0.14). **G, H**. Similar images and analysis for LN/Slit-2 patterns. Note that *TSC2*⁺/⁺ neurites were robustly guided by Slit-2 (*P* < 0.0001), while *TSC2*⁺/⁻ neurites failed to guide (*P* = 0.69). #*P* < 0.0001, Two-tailed Student's *t*-test, represented as mean +/− SEM. Scale, 500 μm (**A**), 100 μm (**C–G**).

required for response to this cue in other model systems[28,29]. Indeed, we observed that control neurons showed no preference on unclustered ephrin-A5 (Fig. 2E, F), confirming that ephrin protein itself was responsible, rather than discontinuous LN or Fc conjugate.

To test whether other canonical cues require TSC2 to guide cortical axons, we tested Netrin-1, a growth-promoting cue, as well as Slit-2, which is another crucial inhibitory axon guidance cue for neural circuit assembly using patterned substrata[30]. Similar to ephrin-A, corrected *TSC2*⁺/⁺ axons are guided along with stripes of pure LN and are robustly repelled from stripes of Slit-2 (Fig. 2G, H). However, as with our findings with ephrin-A, *TSC2*⁺/⁻ axons did not discriminate between Slit-2 and LN stripes and covered patterned substrata equally (Fig. 2G, H). On the other hand, both *TSC2*⁺/⁺ and *TSC2*⁺/⁻ axons were guided by Netrin-1 stripes (Supplementary Fig. 2C, D), suggesting heterozygous loss of TSC2 is not sufficient to cause guidance errors on Netrin-1. Thus, in this model human axon guidance errors by neurons with ASD-related pathogenic mutations can be rescued by gene correction.

**Basal rates of neurite outgrowth are enhanced in *TSC2* patient cortical neurons**. In pathfinding assays, we noted that *TSC2*⁺/⁻ axons appeared longer than corrected control axons (Fig. 2A, B). Therefore, we directly tested how partial and complete loss of TSC2 expression affected neurite outgrowth in fixed neurosphere cultures on homogenous LN substrata. Culturing neurons within neurospheres is advantageous, as greater numbers of neurons generate axons compared to dissociated neurons with significantly less cell death[31]. hFB neurospheres were plated and allowed to grow for 24 h and then fixed and immunostained for tubulin to assay neurite lengths. Neurite extension is profuse and rapid from neurospheres, as some neurites were observed to extend more than one millimeter after 1DIV (Fig. 3A). At 24 h the twenty longest neurites projecting from each sphere were measured and compared between cell lines (Fig. 3B). hFB *TSC2*⁺/⁻ neurites were found to grow significantly longer than multiple lines of *TSC2*⁺/⁺ neurons, including isogenic control neurons. *TSC2*⁻/⁻ axons were also longer than control *TSC2*⁺/⁺ axons, although null neurons were significantly more variable between spheres (F-test, *P* < 0.0001,

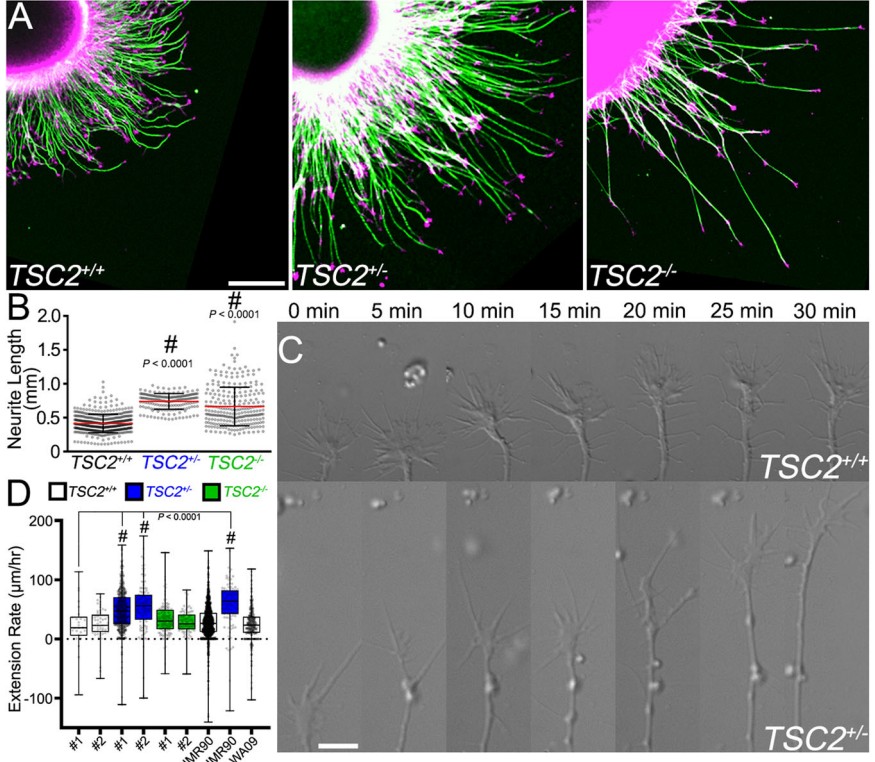

**Fig. 3** *TSC2* **patient cortical neurons exhibit enhanced neurite extension compared to isogenic control neurons. A** hFB neurospheres of indicated genotype after one day in vitro immunolabeled for Acetylated-tubulin (green) and counter-stained for F-actin (magenta). Note longer axons extend from both *TSC2+/−* and *TSC2−/−* neurospheres compared to isogenic control neurons. **B** Average neurite lengths of 20 longest axons per neurosphere of each genotype (*TSC2+/+* vs. *TSC2+/−* and *TSC2+/+* vs. *TSC2−/−*, *P* < 0.0001). **C** Live cell, differential interference contrast (DIC) imaging of growing axons from *TSC2+/+* and *TSC2+/−* hFB neurons at 5 min time intervals. **D** Average axon extension rates show that *TSC2+/−* hFB neurites grew markedly faster compared to isogenic control neurons and *TSC2* null neurons, as well as hFB neurons differentiated from unrelated control iPSCs (IMR90) and ESCs (WA09). CRISPR-Cas9-generated IMR90*TSC2+/−* neurites also grew at similar rates as patient-derived neurites (*TSC2+/+* clone 1 vs. *TSC2+/−* clone 1, *TSC2+/+* clone 1 vs. *TSC+/−* clone 2, *TSC2+/+* clone 1 vs. IMR90*TSC2+/−*, IMR90*TSC2+/−* vs. IMR90*TSC2+/+*, IMR90*TSC2+/−* vs. WA09, *P* < 0.0001). #*P* < 0.0001, One-way ANOVA with Tukey's Multiple Comparison, represented as mean +/− SEM (**B**), and min, max, median, and IQR (D). Scale, 100 μm (**A**), 5 μm (**C**).

*n* = 900) (Fig. 3A, B). Neurite lengths of human motor neurons (hMNs) differentiated from *TSC2+/−* iPSCs were also significantly longer than control neurons (Supplemental Fig. 3C, D).

To confirm these results, we measured the rates of neurite outgrowth by time-lapse imaging of live hFB neurons from each genotype, which eliminates potential bias that may result from measuring the longest axons in our previous analysis. Representative differential interference contrast (DIC) images of live *TSC2+/+* and *TSC2+/−* growth cones over thirty-minute time periods show that *TSC2+/−* axons extend nearly two times faster than control neurons from several different iPSC lines (Fig. 3C). Consistent with increased variability of fixed *TSC2−/−* neurite lengths, the average rate of outgrowth by null neurons was more variable and on average slower compared to *TSC2+/−* neurons. Similar results were obtained with hMNs (Supplemental Fig. 3E). Note that these observations were consistent across passage number, and independent of neurosphere sizes, which were between 200 and 1000 microns in diameter (Supplemental Fig. 4A, B).

***TSC2+/−* neurons are less sensitive to inhibitory axon guidance cues**. While we showed that TSC2 was necessary for hFB neuron guidance in fixed stripe assays (Fig. 2), it is unknown whether TSC2 is required for an acute sensitivity to guidance cues, which may involve local regulation of protein synthesis[18]. We initially tested control and *TSC2+/−* cortical neuron sensitivities to bath

applied ephrin-A1. Corrected *TSC2+/+* control neurons, as well as unrelated control neurons (Supplemental Fig. 4C), rapidly and robustly collapsed in response to 2 μg/ml ephrin-A1 (high dose) (Fig. 4A). Most control growth cones collapsed, and many axons retracted in response to high dose ephrin-A1 (Fig. 4B), while a ten-fold lower dose of ephrin-A1 (0.2 μg/ml) had more modest effects on extension rates (Fig. 4C). In contrast, *TSC2+/−* hFB neurites were modestly inhibited by high dose ephrin-A1 (Fig. 4A–C) and showed little change in extension rate upon exposure to a low dose of ephrin-A1 (Fig. 4C). While control growth cones robustly retracted in response to a high dose of ephrin-A1, *TSC2+/−* neurons were frequently observed to transiently pause or show mild growth cone area reduction without retracting (Fig. 4A, B). The dose-dependent effect of ephrin-A1 suggests that *TSC2+/−* growth cones retain EphA receptors, which we confirmed to be at similar expression levels by immunolabeling for EphA2 and EphA4 (Supplemental Fig. 5A–D). The extent of growth cone collapse was also quantified in hFBs and hMNs 15 min after ephrin-A1 treatment (see "Methods" section). Consistent with time-lapse imaging, we found that both isogenic control and unrelated control hFB neurons (Fig. 4D–F, Supplemental Fig. 4C), as well as hMNs (Supplemental Fig. 3F), were significantly more sensitive to ephrin-A1 compared to *TSC2+/−* neurons.

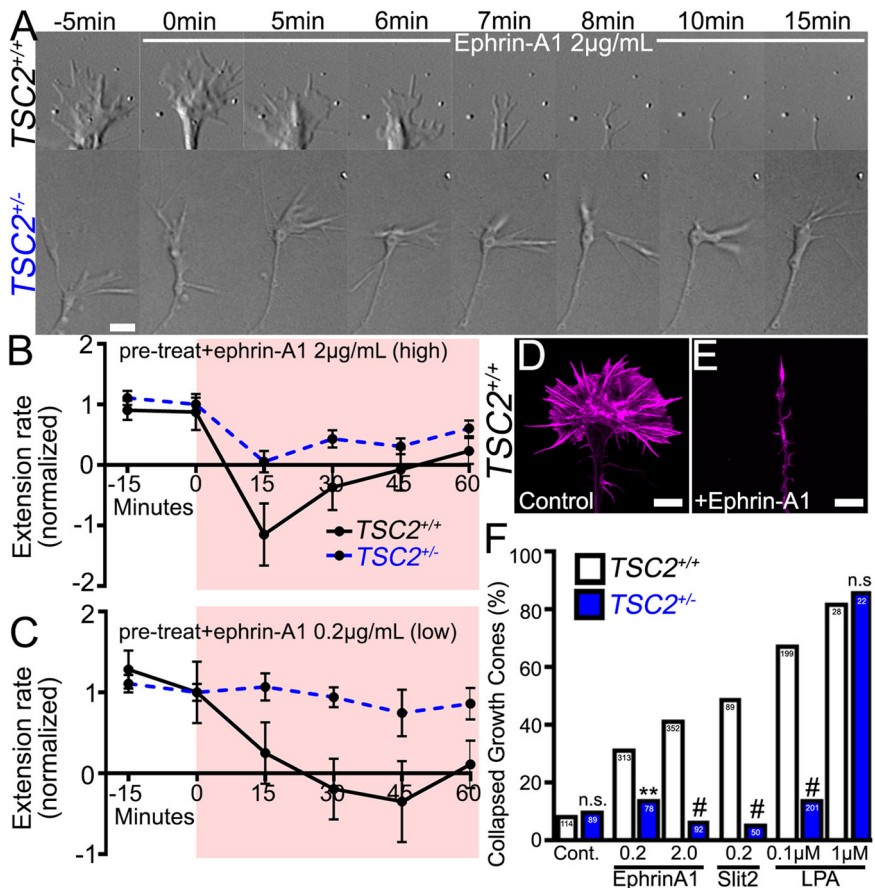

**Fig. 4 $TSC2^{+/-}$ neurites are less sensitive to inhibitory guidance factors. A** $TSC2^{+/+}$ and $TSC2^{+/-}$ hFB neurons were acutely treated with ephrin-A1 (2 µg/mL) over indicated time. Note the rapid collapse of the $TSC2^{+/+}$ growth cone, but only pausing of the $TSC2^{+/-}$ growth cone. **B**, **C**. Neurite extension rates normalized to pre-application extension rates for high (**B**) and low dose ephrin-A1 (**C**). In response to 2 µg/ml ephrin-A1, $TSC2^{+/+}$ neurites retracted within minutes and rarely restored outgrowth, while $TSC2^{+/-}$ neurites only transiently paused extension. In response to 0.2 µg/ml ephrin-A1, $TSC2^{+/+}$ neurites slowly retracted while $TSC2^{+/-}$ neurites showed little effect. **D**, **E** F-actin labeled $TSC2^{+/+}$ hFB growth cones fixed 15 min after control treatment (**D**) or 2 µg/ml ephrin-A1 (**E**). **F** Analysis of percent growth cone collapse by $TSC2^{+/+}$ and $TSC2^{+/-}$ hFB neurons in response to ephrin-A1 (media control: $TSC2^{+/+}$ vs. $TSC2^{+/-}$, $P = 0.811$; low dose: $TSC2^{+/+}$ vs. $TSC2^{+/-}$, $P = 0.018$; high dose: $TSC2^{+/+}$ vs. $TSC2^{-/-}$, $P < 0.0001$) and Slit-2 (in µg/ml) ($TSC2^{+/+}$ vs. $TSC2^{-/-}$, $P < 0.0001$), as well as LPA (in µM) (low dose: $TSC2^{+/+}$ vs. $TSC2^{+/-}$, $P < 0.0001$; high dose: $TSC2^{+/+}$ vs. $TSC2^{-/-}$, $P = 0.999$). Control growth cones collapsed as in (**E**) approximately 40% of the time (352 growth cones imaged across 26 neurospheres). $TSC2^{+/-}$ growth cones failed to collapse in response to the repulsive cue Slit-2 or to a low dose of the GPCR ligand and RHOA pathway activator lysophosphatidic acid (LPA) (100 nM). A higher dose of LPA (1 µM) induced collapse in $TSC2^{+/-}$ growth cones. **$P < 0.01$, #$P < 0.0001$, Two-tailed Fisher's Exact Test. Scale, 5 µm.

To determine whether the observed insensitivity of $TSC2^{+/-}$ growth cones was unique to ephrin-A1, we assayed additional canonical inhibitory cues (Fig. 4F and Supplemental Fig. 4C). As with ephrin-A1, hSlit2 (200 ng/mL) elicited a robust collapse response in control hFB neurons but had little effect on $TSC2^{+/-}$ neurons. Similarly, lysophosphatidic acid (LPA), a G-protein coupled receptor-ligand and potent activator of RHOA strongly collapsed control neurons at 100 nM but had no effect on $TSC2^{+/-}$ neurons at this dose. However, 1 µM LPA did collapse $TSC2^{+/-}$ growth cones, suggesting that these mutant neurons are intrinsically capable of collapsing, but are less sensitive to multiple inhibitory ligands (Fig. 4F, Supplementary Fig. 4C). This latter result is in agreement with previous work in a mouse model showing that $Tsc2^{+/-}$ neurons were sensitive to a high dose of LPA[18]. Finally, while control hMNs were highly sensitive to Semaphorin-3F, $TSC2^{+/-}$ hMN growth cones were less sensitive (Supplemental Fig. 3F). Together these results show that haploinsufficient TSC2 human neurons are less sensitive to several different types of inhibitory cues, suggesting that TSC2 may mediate signaling downstream of diverse receptors.

**Ephrin-mediated changes in local protein synthesis are unaffected in $TSC2^{+/-}$ neurons.** In rodent neurons, ephrin-mediated growth cone collapse is correlated with inhibition of protein synthesis[18]. To test whether similar mechanisms may operate in human neurons, we first assayed protein synthesis within growth cones of control and TSC2 patient neurons. Given that control and $TSC2^{+/-}$ growth cones showed similar levels of MTOR activity (Fig. 1C–E, Supplementary Fig. 1G), we hypothesized that constitutive translation would not differ between these groups but may be elevated in $TSC2^{-/-}$ neurons. To assess rates of protein synthesis within growth cones we utilized the SUnSET technique, which uses an anti-puromycin antibody to detect puromycin-labeled nascent peptides[32] and has been used successfully to assay LPS in growth cones[33]. Briefly, the antibiotic puromycin serves as a cell-permeable tRNA analog that binds translating polypeptides but does not substantially impact general translation when used at a low dose. Initially, we assessed whether the basal rates of protein synthesis differed between genotypes by adding puromycin to cultured neurons for ten minutes to provide a readout of local translation within the growth cones. Short puromycin incubation times using neurons that were at a minimum of 300

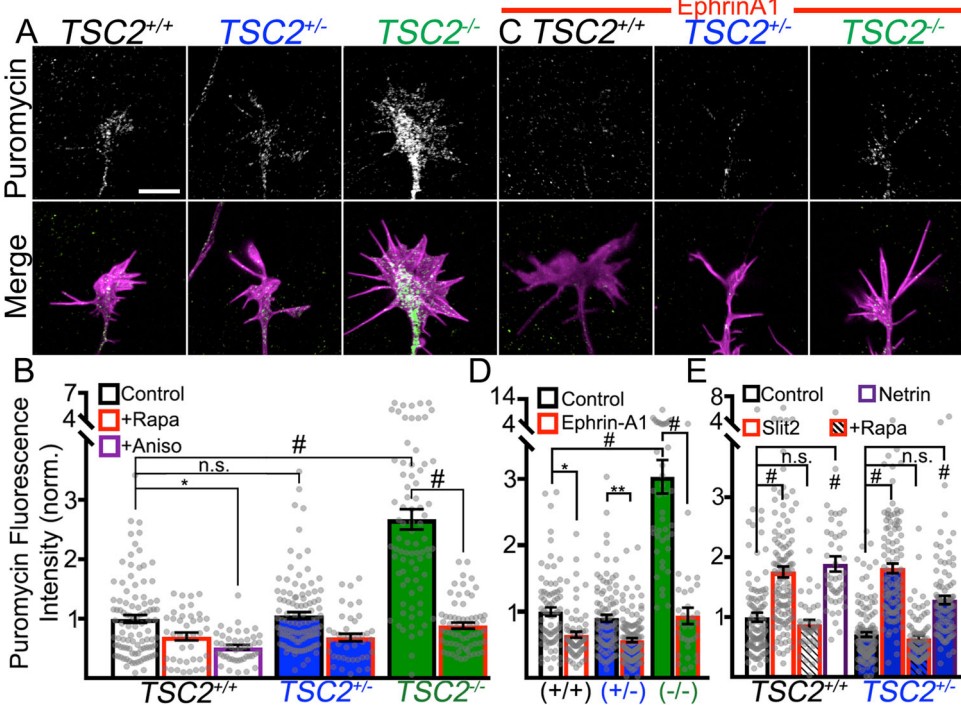

**Fig. 5 $TSC2^{+/+}$ and $TSC2^{+/-}$ cortical growth cones modulate protein synthesis in response to cues. A** Puromycin labeled growth cones (green in merges) of indicated genotypes counter-stained for F-actin (magenta in merges). Puromycin-labeled proteins were detected with anti-puromycin antibodies (see "Methods" section). **B** Normalized fluorescence intensities (to untreated control growth cones) show that basal local protein synthesis (LPS) rates were not significantly different between $TSC2^{+/+}$ and $TSC2^{+/-}$ hFB growth cones ($P = 0.999$), while $TSC2^{-/-}$ growth cones show a marked increase in LPS ($TSC2^{+/+}$ vs. $TSC2^{-/-}$, $P < 0.0001$). LPS was inhibited with 30 min pre-incubation of anisomycin (40 nM) ($P = 0.015$), and rapamycin incubation restored the rate of LPS by $TSC2^{-/-}$ neurons down to control levels (Rapa groups: $TSC2^{+/+}$, $P = 0.40$; $TSC2^{+/-}$, $P = 0.08$; $TSC2^{-/-}$, $P < 0.0001$). **C, D** LPS within the growth cones was significantly reduced by EphrinA1 (2 µg/mL) treatment in all genotypes (Ephrin-A1: $TSC2^{+/+}$, $P = 0.028$; $TSC2^{+/-}$, $P = 0.007$; $TSC2^{-/-}$, $P < 0.0001$). **E** Ten min. treatment with Slit-2 (200 ng/mL) and netrin-1 (100 ng/mL) increased PS in $TSC2^{+/+}$ and $TSC2^{+/-}$ hFB growth cones (control vs. Slit-2: $TSC2^{+/+}$, $P < 0.0001$; $TSC2^{+/-}$, $P < 0.0001$; control vs. netrin-1: $TSC2^{+/+}$, $P < 0.0001$; $TSC2^{+/-}$, $P < 0.0001$). Slit-2-mediated LPS was sensitive to rapamycin in both $TSC2^{+/+}$ and $TSC2^{+/-}$ hFB growth cones (control vs. Slit-2 + rapamycin: $TSC2^{+/+}$, $P = 0.985$; $TSC2^{+/-}$, $P = 0.999$). One-way ANOVA with Tukey's Multiple Comparison, represented as mean +/− SEM. *$P < 0.05$, **$P < 0.001$, #$P < 0.0001$. Scale, 5 µm.

microns from their cell bodies, ensured that the nascent proteins we measured were locally synthesized within growth cones. As hypothesized, the rate of local translation was not different between control and $TSC2^{+/-}$ growth cones (Fig. 5A, B). However, in agreement with our ribosomal protein activation results (Fig. 1C–E), constitutive translation was markedly upregulated in $TSC2^{-/-}$ growth cones. This suggests that under basal conditions mTOR-dependent protein synthesis is unaffected in $TSC2^{+/-}$ growth cones. To verify that mTOR regulates LPS in hFB growth cones, we treated neurons with rapamycin, which decreased basal rates of LPS within control and $TSC2^{+/-}$ growth cones and normalized LPS within $TSC2^{-/-}$ growth cones (Fig. 5B). As an additional control, co-treatment with the protein synthesis inhibitor anisomycin also decreased basal LPS within control growth cones (Fig. 5B).

Several groups have reported in amphibian and rodent animal models that a variety of axon guidance cues affect growth cone motility and guidance by modulating mTOR-dependent LPS[18,34,35], but similar findings have not been reported in human neurons. To first test whether guidance cues regulate LPS in human neuronal growth cones, we exposed control growth cones to puromycin at defined intervals before and after cue stimulation (Fig. 5C–E). EphrinA1 has been shown to decrease protein synthesis in a variety of cell types, including developing neurites and growth cones[18,36]. We found this to be true in human cortical neurons as well, as ephrin-treated growth cones showed significantly reduced puromycin incorporation (Fig. 5C, D).

Since we found that $TSC2^{+/-}$ neurons only weakly respond to several cues, we considered that mTOR-dependent regulation of LPS in response to these cues may be defective, even though basal p-S6 (Fig. 1C–E, Supplemental Fig. 1G) and rates of protein synthesis are unaffected (Fig. 5A, B). However, ephrin-A1 treatment reduced protein synthesis in $TSC2^{+/-}$ growth cones to a similar degree (Fig. 5C, D). More surprising, ephrin-A1 treatment also decreased puromycin incorporation in $TSC2^{-/-}$ growth cones, suggesting that protein synthesis regulation in response to this cue is independent of TSC2. While Slit2 stimulation also leads to growth cone collapse (Fig. 4D, Supplemental Fig. 4), in contrast to ephrins, Slit2 increases protein synthesis in growth cones[35], which we confirmed in human neurons (Fig. 5E). However, while $TSC2^{+/-}$ growth cones do not collapse in response to Slit2 (Fig. 4F), they still exhibit increased protein synthesis in response to Slit2 (Fig. 5E). To test whether LPS in response to Slit2 required mTOR, we co-treated neurons with rapamycin, which suppressed the increase in protein synthesis in both $TSC2^{+/+}$ and $TSC2^{+/-}$ neurons (Fig. 5E and Supplemental Fig. 6), suggesting that while MTOR promotes LPS in response to Slit-2, this pathway is normal in a TSC2 haploinsufficiency background. We also found that $TSC2^{+/+}$ and $TSC2^{+/-}$ neurons increased protein synthesis in response to Netrin-1 (Fig. 5E, Supplemental Fig. 6), a growth-promoting guidance cue, in agreement with previous work[34,37,38]. Together, these results suggest that growth cones modulate LPS in response to guidance cues and that this modulation is grossly normal in

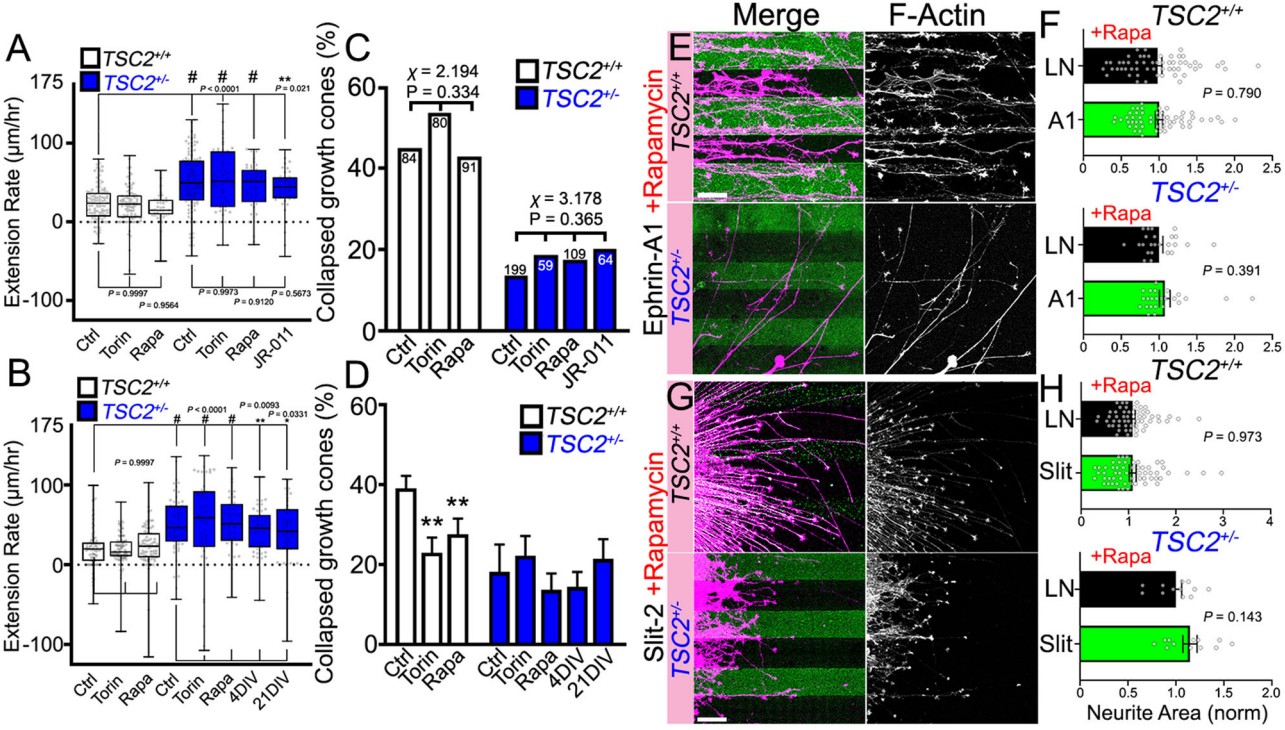

**Fig. 6 The effects of TSC2 loss of function are independent of mTORC1 and mTORC2. A**, **B** Axon extension rates of $TSC2^{+/+}$ and $TSC2^{+/-}$ hFB neurons were not significantly affected by mTORC1 or mTORC2 inhibitors that were applied acutely (**A**, 1 h) (control vs. torin-1: $TSC2^{+/+}$ $P = 0.999$; $TSC2^{+/-}$ $P = 0.997$; control vs. rapamycin: $TSC2^{+/+}$ $P = 0.956$; $TSC2^{+/-}$ $P = 0.913$; control vs. JR-AB2-011: $TSC2^{+/-}$ $P = 0.561$) or chronically (**B**, 24 h or as indicated) (control vs. torin-1: $TSC2^{+/+}$ $P = 0.999$; $TSC2^{+/-}$ $P = 0.999$; control vs. rapamycin: $TSC2^{+/+}$ $P = 0.993$; $TSC2^{+/-}$ $P = 0.999$; control vs. rapamycin 4DIV: $TSC2^{+/-}$ $P = 0.978$, control vs. rapamycin 21DIV: $TSC2^{+/-}$ $P = 0.840$). mTORC1 was specifically inhibited with acute rapamycin (40 nM), mTORC1/C2 were both inhibited with torin-1 (100 nM), and mTORC2 was specifically inhibited with JR-AB2-011 (1 μM). **C** Acute MTOR modulation with 30 min pre-incubation had no effect on collapse response of $TSC2^{+/+}$ growth cones ($P = 0.334$, $\chi^2 = 2.194$, $df = 2$) and failed to rescue cue response in $TSC2^{+/-}$ growth cones ($P = 0.365$, $\chi^2 = 3.178$, $df = 3$). **D** Chronic (24 h) mTOR inhibition with rapamycin (40 nM) or torin-1 (100 nM) reduced the sensitivity of $TSC2^{+/+}$ growth cones to ephrin-A1 ($P = 0.012$, $\chi^2 = 8.801$, $df = 2$) but did not rescue desensitized $TSC2^{+/-}$ growth cones. Long-term treatment with low dose rapamycin (5 nM) throughout differentiation also failed to modulate the collapse response of $TSC2^{+/-}$ neurons ($P = 0.536$, $\chi^2 = 3.131$, $df = 4$). **E**–**H** Analysis of thresholded area occupied by neurites (F-actin channel) normalized to the area of neurites growing on laminin only for ephrin-A1 (**E**, **F**) and Slit-2 patterns (**G**, **H**), as described previously. Inhibition of MTOR with rapamycin (40 nM) blocked guidance of $TSC2^{+/+}$ neurites on patterned ephrin-A1 (10 μg/mL) ($P = 0.790$) and failed to rescue $TSC2^{+/-}$ misguidance ($P = 0.391$) (**E**, **F**). Guidance of $TSC2^{+/+}$ on patterned Slit-2 was blocked by chronic MTOR inhibition with rapamycin ($P = 0.973$) and rapamycin failed to rescue $TSC2^{+/-}$ misguidance ($P = 0.143$) (**G**, **H**). One-way ANOVA with Tukey's Multiple Comparison (**A**–**B**), chi-squared test (**C**–**D**), or Two-tailed Student's *t*-test (**F** and **H**), data represented as min, max, median, and IQR (**A**, **B**), and mean +/− SEM (**F**, **H**). *$P < 0.01$, **$P < 0.01$, #$P < 0.0001$. Scale, 100 μm.

$TSC2^{+/-}$ growth cones. These results also indicate that changes in LPS are not necessary for the acute effects of these guidance cues.

**TSC2 functions independently of MTOR to regulate axon extension and cue sensitivity.** As we found that MTOR-mediated LPS is normal in $TSC2^{+/-}$ growth cones, we sought to investigate whether enhanced neurite extension by $TSC2^{+/-}$ neurons may not involve MTOR. Using mTORC1-specific and mTORC2-specific, as well as dual inhibitors, we first tested the effects of MTOR inhibition on axon extension. To our surprise, acute pharmacological inhibition of mTORC1 with rapamycin had little effect on axon extension rates of $TSC2^{+/+}$ or $TSC2^{+/-}$ neurons (Fig. 6A). As previous reports have also implicated mTORC2 upstream of cytoskeletal dynamics in other cell types[39], we tested the mTORC1/C2 inhibitor torin-1[40] and the mTORC2-specific inhibitor JR-AB2-011[41]. Again, neither of these manipulations significantly affected basal outgrowth of $TSC2^{+/+}$ neurites nor enhanced outgrowth of $TSC2^{+/-}$ neurites (Fig. 6A). In addition, we tested whether longer treatment with mTORC1 or mTORC2 inhibitors normalized $TSC2^{+/-}$ neurite outgrowth since TSC2 haploinsufficiency is known to induce proteomic changes[23,42]. Again, we found that neither 24 h treatment with rapamycin or

torin-1, nor long-term treatment with rapamycin for up to 21 days during neuronal differentiation, reversed the enhanced extension rates exhibited by $TSC2^{+/-}$ neurons (Fig. 6B). We confirmed that these pharmacological treatments effectively inhibited mTORC1 and mTORC2 in growth cones (Supplemental Fig. 7A, B). These data are consistent with the normal levels of S6 phosphorylation observed in $TSC2^{+/-}$ growth cones (Fig. 1C–E, Supplemental Fig. 1G), and with some rodent and human studies[24,25]. Together, these surprising findings suggest that TSC2 has MTOR-independent targets within growth cones that constrain neurite outgrowth since TSC2 haploinsufficiency results in enhanced axon extension.

To determine if either mTORC1 or mTORC2 complexes function downstream of TSC2 to regulate guidance cue sensitivity, we tested the effects of pharmacological inhibitors of each mTOR complex on ephrin-mediated growth cone collapse. Consistent with the results described above, we found that acute treatment of $TSC2^{+/+}$ or $TSC2^{+/-}$ hFB neurons with mTORC1/C2 inhibitors neither prevents, nor restores ephrinA1-mediated collapse by control or heterozygous TSC2 patient neurons, respectively (Fig. 6C). These results suggest that acute signaling downstream of EphA does not require mTORC1/C2 activity.

However, while pre-treatment of hFB neurons with mTOR inhibitors for 24 h before stimulation with ephrin-A1 does partially block ephrinA1 collapse of $TSC2^{+/+}$ hFB axons, it does not restore sensitivity of $TSC2^{+/-}$ hFB neurons (Fig. 6D). Loss of sensitivity of control neurons to ephrin-A1 may be due to mTORC1-dependent proteomic changes, but this appears unrelated to changes caused by loss of TSC2 function. These surprising results suggest that hyperactive mTORC1/C2 signaling does not account for the phenotypes we observe in developing $TSC2^{+/-}$ axons.

Finally, we used pharmacologic manipulations as described previously to test whether mTORC1/C2 inhibition could disrupt hFB axon guidance. Stripe assays were performed as described (Fig. 2), with the addition of rapamycin at the time of cell culture. Note that chronic rapamycin treatment inhibits both mTORC1 and mTORC2[43,44]. Consistent with our results with collapse assays (Fig. 6D), we found that chronic rapamycin treatment prevented guidance of control $TSC2^{+/+}$ hFB neurons on ephrin-A1 and Slit-2 stripes (Fig. 6E, F), suggesting either that active MTOR is necessary for more sensitive guidance behaviors or that chronic proteomic changes disrupt guidance. Importantly, rapamycin treatment did not restore guidance of $TSC2^{+/-}$ neurons on patterned ephrin-A1 or Slit-2 (Fig. 6E, F), suggesting that misguidance is not due to hyper-active mTORC1/C2. Together, these findings indicate that chronic mTORC1 activity is likely necessary downstream of ephrin-A, Slit-2, and netrin-1, but that TSC2 functions independently of mTORC1 to regulate guidance by these cues.

**RHOA signaling is reduced in $TSC2^{+/-}$ patient neurons and growth cones**. Considering our results showing that basal neurite extension rates and guidance cue sensitivity are abnormal in $TSC2^{+/-}$ neurites, while MTOR signaling appears unaffected, we looked to other possible downstream targets of TSC2. Studies in $TSC2^{-/-}$ cells have shown that RHOA, a Ras family small G protein, is regulated downstream of TSC1/TSC2 independent of mTOR complexes[26,45]. RHOA controls F-actin contractility within growth cones, which modulates both axon extension rate[30,46] and is required for the collapse in response to several inhibitory axon guidance cues, including ephrin-A, Slit2, Semaphorins, and LPA[4,30,47]. To begin to test whether basal RHOA signals are altered in $TSC2^{+/-}$ growth cones, we first used a RHOA fluorescence resonance energy transfer (FRET) biosensor[48] to measure RHOA activity within $TSC2^{+/+}$ and $TSC2^{+/-}$ hFB growth cones. By performing acceptor photobleaching (i.e., donor-dequenching)[49], we found that FRET efficiency was lower in $TSC2^{+/-}$ compared to $TSC2^{+/+}$ growth cones, indicating lower baseline RHOA activity (Fig. 7A), which may account for faster neurite extension rates by $TSC2^{+/-}$ neurons (Fig. 3). To corroborate FRET results, we measured active RHOA from whole neurosphere cell lysates using a modified enzyme-linked immunosorbent assay (ELISA) for active RHOA (G-LISA, Cytoskeleton) and found that active RHOA also appeared lower in $TSC2^{+/-}$ neurospheres compared to control neurospheres (Fig. 7C). However, while basal RHOA activity was only modestly lower in $TSC2^{+/-}$ neurons, differences were more pronounced in response to ephrin-A1 treatment. Live neurosphere cultures were lysed after 5 min treatment with 2 μg/ml ephrin-A1 and RHOA was measured using G-LISA. Here we found that control $TSC2^{+/+}$ neurons showed robust activation of RHOA, while $TSC2^{+/-}$ neurons showed no activation (Fig. 7B). Total RHOA expression appeared to be similar across neurospheres from each genotype (Supplementary Fig. 8C, D). These results suggest that biallelic expression of TSC2 is necessary for proper RHOA activation in human neurons downstream of ephrin-A1.

RHOA promotes actomyosin contractility by activating Rho-associated kinase (ROCK), which phosphorylates myosin light chain (MLC) directly and inactivates myosin light chain phosphatase (MLCP) to further elevate phospho-MLC (p-MLC)[50]. Therefore, the intensity of p-MLC immunolabeling provides a useful readout of RHOA activity and the strength of actomyosin contractility. Consistent with RHOA measurements, we found that basal p-MLC immunolabeling was lower in $TSC2^{+/-}$ compared to $TSC2^{+/+}$ growth cones (Fig. 7D, F). To determine if TSC2 is necessary for acute activation of MLC by ephrin-A1, we treated $TSC2^{+/+}$ and $TSC2^{+/-}$ hFB neurons for 5 min with 2 μg/ml Ephrin-A1 prior to immunolabeling. Similar to RHOA activity measurements (Fig. 7C), we observed strong activation of MLC in $TSC2^{+/+}$ hFB neurons after 5 min treatment with ephrin-A1, but little change in $TSC2^{+/-}$ hFB neurons (Fig. 7E, F).

**RHOA-ROCK-myosin regulates axon extension and guidance downstream of TSC2**. Since RHOA and MLC activity appear lower in $TSC2^{+/-}$ hFB growth cones compared to control (Fig. 7A–D), we hypothesized that inhibition of this pathway may differentially affect axon extension by $TSC2^{+/+}$ compared to $TSC2^{+/-}$ neurons. To test this possibility, we treated neurons with RHOA-ROCK-myosin inhibitors and evaluated the effects on basal axon outgrowth and response to ephrin-A1. Interestingly, acute treatment of control $TSC2^{+/+}$ neurons with either ROCK (Y-27632) or myosin II (blebbistatin) inhibitors stimulated the rate of axon outgrowth to rates comparable with $TSC2^{+/-}$ neurites but had no significant effect on already rapid $TSC2^{+/-}$ neurite outgrowth (Fig. 8A, B). These results suggest that the RHOA/ROCK/myosin II axis may be minimally active in $TSC2^{+/-}$ growth cones so inhibition cannot further lower myosin activity, which is consistent with our direct measurements of this pathway (Fig. 7A–D). To directly activate myosin II to test how this may affect axon extension by $TSC2^{+/+}$ and $TSC2^{+/-}$ neurons, we tested the effects of CalyculinA (CalyA). CalyA is a potent inhibitor of protein phosphatase 1 (PP1), which is the catalytic subunit of myosin light chain phosphatase[51]. Five min treatment with 200 pM CalyA increased p-MLC in $TSC2^{+/+}$ and $TSC2^{+/-}$ growth cones (Supplemental Fig. 8A) and inhibited axon extension of $TSC2^{+/-}$ growth cones (Fig. 8B). To corroborate pharmacological manipulation of the RHOA pathway, we also overexpressed RHOA-WT directly in $TSC2^{+/-}$ neurons via lentiviral transduction. Overexpression of RHOA-WT in $TSC2^{+/-}$ growth cones reduced outgrowth rates and increased MLC activation compared to RFP-control (Fig. 8B, Supplementary Fig. 8E–F). Taken together, these data suggest that MLC-mediated actomyosin contractility can be rescued in TSC2 patient neurons and is sufficient to slow axon extension.

Next, we tested how blocking the RHOA/ROCK/myosin II pathway affected ephrin-A1-induced collapse in $TSC2^{+/+}$ compared to $TSC2^{+/-}$ neurons. Pre-treatment of control hFB neurons with RHOA, ROCK, or myosin II inhibitors prevented ephrin-A1 dependent collapse, confirming that ephrin-A1 signals through the RHOA pathway in human neurons (Fig. 8C). However, $TSC2^{+/-}$ neurons, which are largely insensitive to ephrin-A1, are not further inhibited by blocking the RHOA pathway, suggesting this pathway is already largely inactive in $TSC2^{+/-}$ neurons (Fig. 8C). In contrast, activation of MLC with low dose CalyA, which slows outgrowth (Fig. 8B), but does not collapse growth cones (Supplementary Fig. 8B), sensitizes $TSC2^{+/-}$ neurons to ephrin-A1 mediated collapse (Fig. 8C). RHOA-WT overexpression within $TSC2^{+/-}$ neurons is also sufficient to rescue growth cone collapse in response to ephrin-A1 (Fig. 8C).

Finally, we tested whether RHOA signaling was necessary for guidance on ephrin-A1 stripes. $TSC2^{+/+}$ and $TSC2^{+/-}$ neurons were cultured in the presence of Y-27632 on stripes of ephrin-A1. Control neurons with ROCK inhibited showed no significant

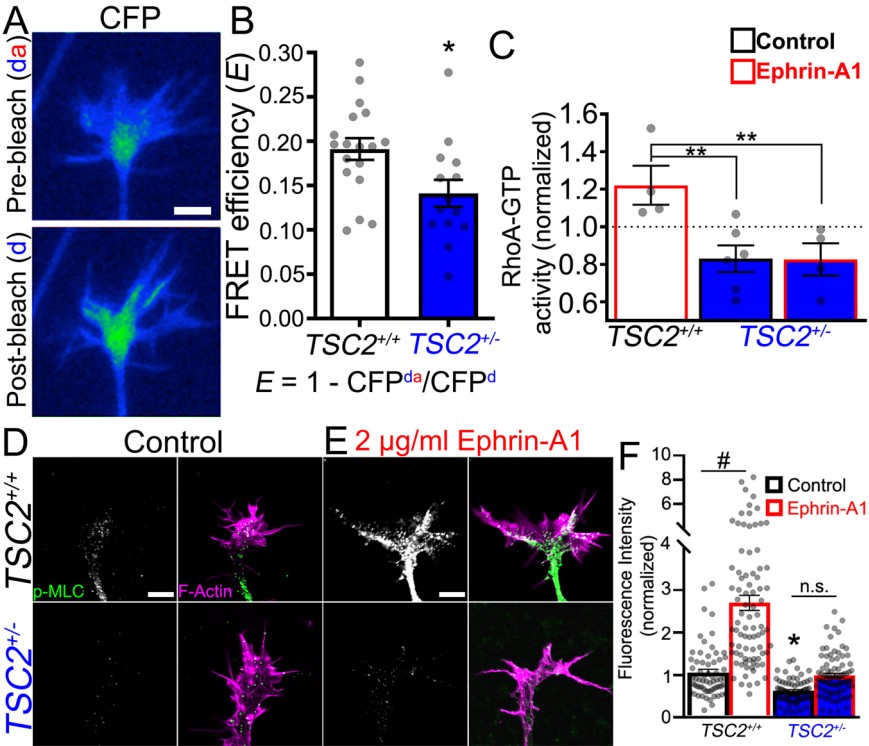

**Fig. 7 *TSC2*⁺/⁻ neurons have reduced RHOA-ROCK-MLC signaling. A** CFP (donor) fluorescent images of RHOA FRET sensor-expressing control hFB growth cone before (above) and after YFP (acceptor) bleaching. **B** Analysis of FRET efficiency in *TSC*⁺/⁺ compared to *TSC*⁺/⁻ growth cones (P = 0.0148). **C** Analysis of RHOA activity by G-LISA in whole *TSC2*⁺/⁺ and *TSC2*⁺/⁻ neurospheres treated with Ephrin-A1. RHOA activity was normalized to untreated control neurons. Active RHOA was significantly increased in *TSC2*⁺/⁺ neurons in response to Ephrin-A1 (2 µg/mL) compared to *TSC2*⁺/⁻ neurons (ephrin-A1 vs. ephrin-A1: P = 0.0099, *TSC2*⁺/⁺ ephrin-A1 vs. *TSC2*⁺/⁻ control: P = 0.0053) Note that basal RHOA activity also trended 20% lower in *TSC2*⁺/⁻ compared to *TSC2*⁺/⁺ neurospheres (control vs. control, P = 0.254). n = 6 independent sets of 30 neurospheres of each genotype for each control group and 4 independent sets of 30 neurospheres for each ephrin-A1 treated group. **D, E** hFB neuronal growth cones of indicated genotype immunolabeled for p-MLC (green in merge) and counterstained for F-actin (magenta in merge) in control (**D**) and after ephrin-A1 stimulation (**E**). Note *TSC2*⁺/⁻ growth cones show significantly lower basal p-MLC labeling relative to control (P = 0.0452). *TSC2*⁺/⁺ neurons treated for 5 min with ephrin-A1 show a robust increase in p-MLC (P < 0.0001), but *TSC2*⁺/⁻ growth cones show little change in p-MLC in response to ephrin-A1 (P = 0.0910). **F** Fluorescence intensity measurements normalized to untreated control neurons for each condition. One-way ANOVA with Tukey's Multiple Comparison (**C** and **F**) or Two-tailed Student's *t*-test (**B**), represented as mean +/− SEM. *P < 0.05, **P < 0.01, #P < 0.0001. Scale, 5 µm.

substratum preference (Fig. 8D, E). *TSC2*⁺/⁻ hFB neurites, which normally are not guided by ephrin-A1 (Fig. 2), showed no apparent effects from this treatment (Fig. 8D, E). We next attempted to rescue guidance effects in *TSC2*⁺/⁻ hFB neurites via RHOA pathway activation. While acute calyculinA treatment increased myosin activity and rescued outgrowth and cue response defects, chronic calyculinA treatment often prevented axon outgrowth even at low doses, proving cytotoxic over the course of 3DIV in agreement with previous work[52]. *RHOA-WT* transduction also did not rescue guidance phenotypes, likely due to the low rate of transduction of post-mitotic neurons (Supplemental Fig. 8E) and the preference for individual neurites to fasciculate with other neurites within the 3DIV time course. These data indicate that the RHOA/ROCK/myosin II pathway regulates axon extension and response to ephrin-A1, and loss of RHOA signaling phenocopies defects exhibited by *TSC2*⁺/⁻ neurons.

## Discussion
In this paper, we show that hFB cortical neurons differentiated from TSC patient-derived iPSCs exhibit several severe developmental abnormalities compared to their isogenic control counterparts. Axon extension by hFB neurons with heterozygous loss of *TSC2* is two times faster than control neurons

(Fig. 3) and *TSC2*⁺/⁻ neurons are less sensitive to several inhibitory axon guidance cues (Fig. 4). However, while *TSC2*⁺/⁻ neurons show severe growth and axon guidance defects (Fig. 2), MTOR signaling appears normal in *TSC2*⁺/⁻ neurons but is robustly upregulated in *TSC2*⁻/⁻ neurons (Fig. 1). This result suggests that axon extension phenotypes may be due to MTOR-independent activities downstream of TSC2 (Fig. 6). While MTOR signaling is normal in *TSC2*⁺/⁻ neurons, RHOA-ROCK-mediated actomyosin contractility is reduced in *TSC2*⁺/⁻ growth cones, and activating this pathway rescues outgrowth and ephrin collapse phenotypes (Figs. 7 and 8). Together, our results suggest that TSC patients may have neural network connectivity abnormalities due to MTOR-independent TSC2 misregulation of RHOA signaling.

TSC is a neurodevelopmental disorder characterized by cortical tubers and neural network connectivity defects[10]. While cortical tubers and lesions in other organ systems likely result from hyperactive mTORC1-mediated protein synthesis[53], the effects of TSC2 loss of function causing neuronal connectivity defects are less well understood. However, a number of studies using different animal model systems have established roles for local protein synthesis (LPS) downstream of axon guidance cues[54], suggesting that TSC2 may also regulate LPS within growth cones to control neural network wiring. If true, neural network

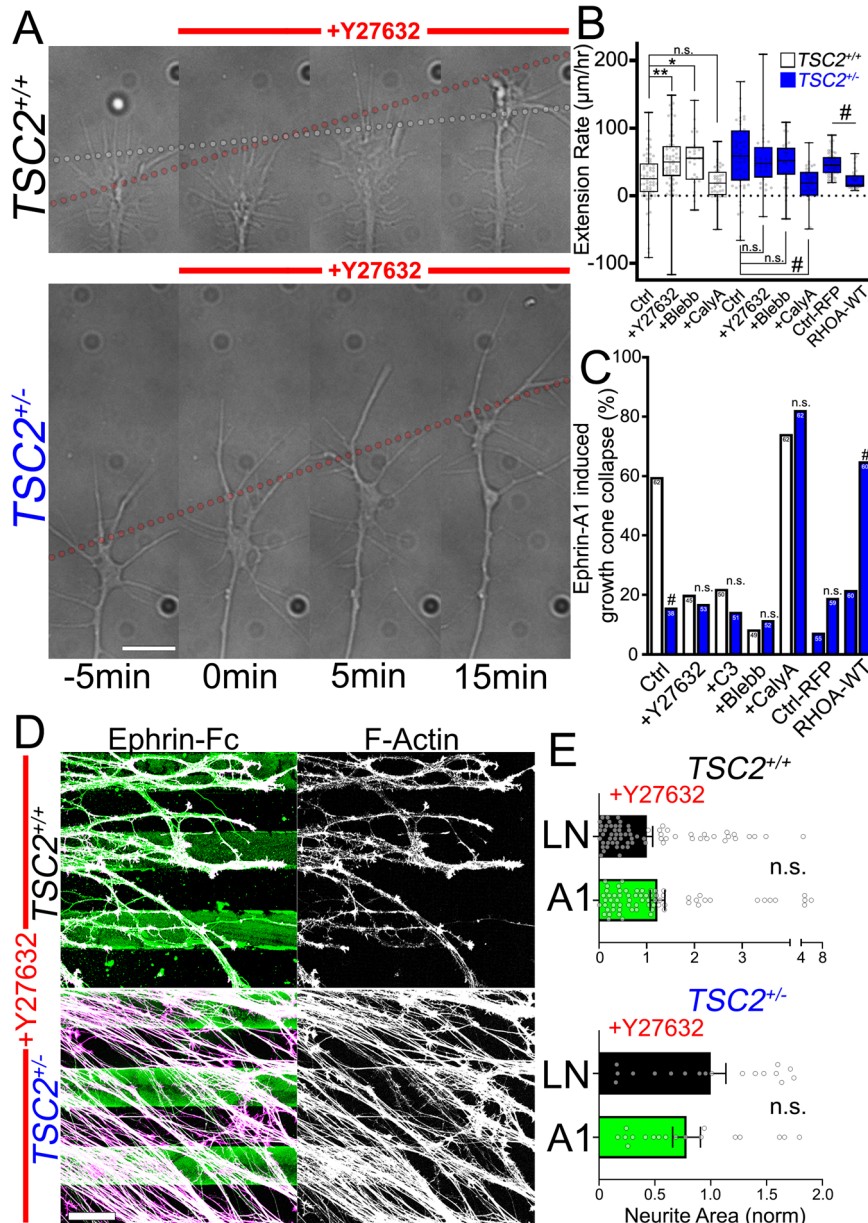

**Fig. 8 RHOA pathway modulation phenocopies and rescues $TSC2^{+/-}$ phenotypes. A** DIC images of growing axons from $TSC2^{+/+}$ and $TSC2^{+/-}$ hFB neurons at 5 min time intervals during treatment with ROCK inhibitor Y-27632 (10 μM). Note that inhibition of ROCK increased $TSC2^{+/+}$ neurite extension rates to a level similar to $TSC2^{+/-}$ neurites, while $TSC2^{+/-}$ neurites did not respond to ROCK inhibitor. Dashed gray line indicates extension rate of control neurons before treatment, while dashed red lines indicate extension rates for $TSC2^{+/-}$ neurons (before and after treatment) and $TSC2^{+/+}$ neurons after treatment. **B**. Neurite extension rate measurements before and after inhibition of ROCK (Y-27632) and myosin-II with blebbistatin (Blebb, 50 μM) in $TSC2^{+/+}$ and $TSC2^{+/-}$ neurons shows $TSC2^{+/+}$ neurite outgrowth accelerated upon ROCK (control vs. ROCK: $TSC2^{+/+}$, $P = 0.0035$; $TSC2^{+/-}$, $P = 0.995$) and myosin-II inhibition (control vs. blebbistatin: $TSC2^{+/+}$, $P = 0.0484$; $TSC2^{+/-}$, $P = 0.971$). Conversely, activation of MLC with the phosphatase inhibitor CalyculinA (200 pM) slowed neurite extension in TSC2$^{+/-}$ neurons to a rate comparable to basal $TSC2^{+/+}$ neurons ($TSC2^{+/+}$ control vs. CalyculinA: $P = 0.996$; $TSC2^{+/+}$ control vs. $TSC2^{+/-}$ CalyculinA: $P = 0.995$; $TSC2^{+/-}$ control vs. $TSC2^{+/-}$ CalyculinA: $P < 0.0001$). Similarly, $TSC2^{+/-}$ neurons over-expressing RHOA-WT-RFP exhibited reduced axon outgrowth rates vs. control-RFP expression ($P < 0.0001$). **C** Ephrin-A1 mediated collapse of $TSC2^{+/+}$ growth cones was prevented by inhibition of RHOA (C3 transferase, 2 μg/mL) (ephrin-A1: $TSC2^{+/+}$ vs. $TSC2^{+/-}$: $P < 0.0001$, ephrin-A1 + C3: $TSC2^{+/+}$ vs. $TSC2^{+/-}$: $P < 0.80$), myosin-II (Blebb, 50 μM) ephrin-A1 + blebbistatin: $TSC2^{+/+}$ vs. $TSC2^{+/-}$: $P < 0.31$), and ROCK (Y-27632, 10 μM) ephrin-A1 + Y-27632: $TSC2^{+/+}$ vs. $TSC2^{+/-}$: $P < 0.74$). Conversely, activation of MLC with a low dose of calyculin (200pM) rescued ephrin-A1 induced collapse by $TSC2^{+/-}$ growth cones (ephrin-A1 + CalyculinA: $TSC2^{+/+}$ vs. $TSC2^{+/-}$: $P < 0.38$), as did RHOA-WT over-expression ($TSC2^{+/-}$ control-RFP control vs. ephrin-A1: $P = 0.0974$; $TSC2^{+/-}$ RHOA-WT control vs. ephrin-A1: $P < 0.0001$). **D** Treatment with Y-27632 blocked $TSC2^{+/+}$ axon guidance on ephrin-A1 stripes ($P = 0.259$) but had no additional effect on $TSC2^{+/-}$ neurons ($P = 0.251$). One-way ANOVA with Tukey's Multiple Comparison (**B**), Two-tailed Fisher's Exact Test (**C**), or Two-tailed Student's $t$-test (**E**), data represented as min, max, median, and IQR (**B**), and mean +/− SEM (**E**). *$P < 0.05$, #$P < 0.0001$. Scale, 5 μm (**A**) and 100 μm (**D**).

abnormalities in TSC patients could be due to dysregulation of MTOR within growth cones. In support of this notion, it was reported in rodent model neurons that activation of TSC2 downstream of ephrin-A decreased mTOR-dependent LPS and that $TSC2^{+/-}$ neurons exhibited axon guidance defects in vivo[18]. Other reports showed that some guidance cues activate mTOR-dependent LPS in growth cones[55], leading us to hypothesize that defects in human neuron axon guidance in TSC patients were due to abnormal regulation of mTOR signaling. However, we were surprised to discover that the axon guidance and outgrowth defects we observed in $TSC2^{+/-}$ neurons were not due to abnormal MTOR signaling.

While recent reports demonstrate that abnormal mTORC1 or mTORC2 signaling may be responsible for developmental defects associated with TSC pathogenic variants in stem cell models[56–60], we find in neuronal growth cones that TSC2 has MTOR-independent targets. Our surprising findings raise the question: What downstream targets of TSC2 are responsible for abnormal axon developmental phenotypes exhibited by human neurons? There are several reports in the literature of mTOR-independent targets of TSC1/TSC2 in a variety of cell types[61]. However, the interpretation of these studies is complicated, as many utilize homozygous null TSC1 or TSC2 cells, which express global proteomic changes from hyperactive mTOR. Nonetheless, potential TSC-associated targets that may affect outgrowth and guidance include FAK[62], HDACs[63], and p53[64], as well as the GTPase RHEB. A recent study utilizing a farnesylation-defective RHEB mutant observed mTORC1-independent enhanced axonal extension rates, consistent with our observations[65]. *WT-Rheb* transfection in mouse RGCs also showed increased resistance to Ephrin-A1-induced collapse, suggesting RHEB may mediate TSC-dependent axon phenotypes[18]. Another potential downstream target of TSC1/2 is the activation of Rho family GTPases, known effectors of neurite outgrowth[66,67]. For example, TSC1 binds Ezrin-Radixin-Moesin (ERM) family proteins and activates RHOA to promote cell adhesion[68]. Consistent with these findings, homozygous knockout of *TSC1* resulted in unstable TSC2 and a decrease in RHOA activity in a mouse fibroblast model[69]. On the other hand, the Krymskaya lab showed that stable transfection of TSC2 upregulated RAC1 at the expense of RHOA activity[26]. This group suggests a model in which TSC1 bound to TSC2 increases RAC1 activity at the expense of RHOA activation. It is possible that TSC2 has cell-type-specific activities, as well as unique targets within neuronal growth cones.

We observed robust phenotypes using heterozygous *TSC2* patient neurons, which is important for several reasons. First, while a minority of cells in some TSC patient tubers may be $TSC2^{-/-}$ due to a second hit mutation, in familial TSC patients with neuropathology, a vast majority of their neurons are $TSC2^{+/-}$, as well as most neurons in TSC patients that acquired early sporadic mutations[12]. Second, complete loss of *TSC2* results in profound MTOR-dependent proteomic changes in cells[23], which confounds mechanistic interpretations. While heterozygous *TSC2* neurons likely also exhibit some MTOR-dependent proteomic differences compared to wild-type neurons, we show that we cannot rescue axon extension or guidance cue sensitivities even with chronic inhibition of MTOR. This result strongly suggests that proteomic changes do not account for the developmental phenotypes we observe in $TSC2^{+/-}$ neurons. However, it is important to note that we do find that chronic MTOR inhibition substantially impacted the ability of normal neurons to be guided in vitro. Given the complex and overlapping roles of MTOR signaling in neurodevelopment, more studies are needed to inform clinical decisions with regard to pharmacological interventions in patient care.

## Methods

**IPSC generation and characterization.** Primary dermal fibroblasts were isolated from tissue acquired with approval from the University of Wisconsin-Madison Human Subjects IRB (protocol #2016–0979) and with patient informed consent. The patient was informed that their participation was entirely voluntary and would not impact their care. Dermal fibroblast samples from normal tissue and an identified hypomelanotic macule (ash leaf spot) were collected from an individual with Tuberous Sclerosis (TSC). The patient presented with high tuber load, epilepsy, autism spectrum disorder, and expressive language delay. Cells were reprogrammed into iPSCs using non-integrating Sendai viruses expressing C-MYC, KLF4, OCT3/4, and SOX2[70]. Lines reprogrammed from each tissue type were verified for the heterozygous nonsense *TSC2* pathogenic mutation at base pair c.972C>G (All primers used in this study are listed in Supplementary Table 2). Characterization was performed utilizing immunofluorescence staining for NANOG, OCT4, and SOX2 with specific antibodies. Mycoplasma testing and karyotyping analysis was provided by WiCell (WiCell, Madison, WI) (Supplemental Fig. 1).

**CRISPR-Cas9 gene editing.** In addition to IMR90 iPS and WA09 (H9) ES cell lines, CRISPR-Cas9 single-strand oligonucleotide (ssODN) method of gene editing was utilized in the heterozygous TSC patient line to correct the nonsense mutation and create isogenic control lines as in Yang, et al. and Chen, et al.[71,72]. A second heterozygous line was similarly engineered in an IMR90 control background. A DNA fragment was inserted into a plasmid downstream of the U6 promoter (Addgene #52961; http://www.addgene.org/52961/) for sgRNA expression targeting exon 9 of *TSC2*. 200 bp homology arms surrounding the mutation site were constructed by Integrated DNA Technologies (IDT). Constructs were electroporated into patient cells followed by puromycin selection, expansion, and sequencing. Off-target analysis was performed via Q5-polymerase PCR for the 5 highest-likelihood off-target sites as predicted by crispr.mit.edu algorithms. All generated lines were verified and characterized as noted above.

**IPSC maintenance.** IPSCs and WA09 (H9) cells were cultured on feeder-free substrate vitronectin (Thermofisher) with Essential 8 Flex media (ThermoFisher) or mTeSR and Matrigel. Cells were passaged 1:18-1:24 every 5–7 days using ReLeSR (Stem Cell Technologies). Pluripotency was confirmed via immunostaining for TRA-1-60 (1:100, ThermoFisher), NANOG (1:200, Cell Signaling Technologies), and OCT4A (1:400, Cell Signaling Technologies). Cells were utilized between passage 18 and 50, and outgrowth analysis confirmed the conservation of neuronal outgrowth phenotypes between early and late passages (Supplementary Fig. 4B).

**Neural differentiation and culture.** hFBs were differentiated as in Doers et al.[73]. At 60–80% confluency, stem cell colonies were lifted with 1 mg/mL Dispase II (Sigma) and cultured in suspension in embryoid body medium (EB) consisting of DMEM/F12 (ThermoFisher), KnockOut Serum (1:4; Thermofisher), MEM NEAA (1:100; ThermoFisher), and Glutamax supplement (1:100; Thermofisher). Over the following three days whole media exchanges were performed with EB and Neural Induction Medium (NIM) consisting of DMEM/F12, MEM NEAA, Glutamax, and heparan sulfate (2 μg/mL; Stem Cell Technologies) in a ratio of 3:1, 1:1, 1:3. On days 4–7 whole media exchanges were made with NIM media. On day 7, cells were plated on laminin (25 μg/mL; Sigma-Aldrich) coated plates and allowed to adhere for 48 h; then half media exchanges of NIM every other day until day 17–18 upon which cells were lifted manually and kept in suspension as neurospheres in Maintenance media consisting of DMEM/F12, MEM NEAA, Glutamax, and B27 supplement minus Vitamin A (1:100; ThermoFisher). Characterization was performed at 10–15DIV via immunostaining for PAX6 (1:20; PAX6 was deposited to the DSHB by Kawakami, A. (DSHB Hybridoma Product PAX6)) and SOX2 (1:100; R&D Systems), and at 25–30DIV via immunostaining for FOXG1 (1:100; Abcam). Motor neurons were differentiated according to established protocols following Du, et al.[74]. Following the day stem cells were passaged, mTeSR media was replaced with Neurobasal:DMEM/F12 at a 1:1 ratio and N2, B27, ascorbic acid (0.1 mM, Santa Cruz), 1× Glutamax, CHIR99021 (3 μM, Tocris), DMH1 (2 μM, Tocris), and SB431542 (2 μM, Tocris). Cells were fed every other until day 7, at which point they were passaged 1:6 and RA (0.1 μM, Stemgent) and Purmorphamine (0.5 μM, Stemgent) were added to the media in addition to reduced CHIR99021 (1 μM) for an additional 6 days with every-other-day feedings. Colonies were lifted and cultured in suspension in neural media with 0.5 μM retinoic acid and 0.1 μM purmorphamine, with every-other-day media changes. Motor neurons were characterized via co-immunostaining for HB9 (1:100, 81.5C10 was deposited to the DSHB by Jessell, T.M./Brenner-Morton, S. (DSHB Hybridoma Product 81.5C10)) and βIII tubulin (1:500, Sigma) (Supplemental Fig. 3A, B).

Initial experiments in differentiation, outgrowth, and collapse were individually analyzed and comparisons were made in a minimum of two lines from the TSC patient, each reprogrammed from a different tissue sample, and two $TSC2^{+/+}$ isogenic controls, showing no significant differences within genotypes. Subsequently, data from multiple clones were pooled for comparisons between $TSC2^{+/-}$ and $TSC2^{+/+}$ isogenic controls. All experiments on post-mitotic neurons

were performed using a minimum of two independent differentiations and four independent neurospheres, except where specifically noted (Supplemental Table 1). IMR90 control and IMR90$^{TSC2+/-}$ cell lines were utilized in key experiments to support the generalizability of key results. Individual numbers of neurospheres, independent lines, and growth cones utilized in each experiment are shown in Supplemental Table 1. Neurons were cultured as neurospheres on acid-washed coverslips coated with poly-D-lysine (PDL, 50 μg/mL; Sigma-Aldrich) and laminin (LN, 25 μg/mL; Sigma-Aldrich) in Neurobasal (ThermoFisher) media with B27 supplement and Glutamax (NB) for 1-14DIV with half media changes every third day as needed. For experiments with dissociated neurons, neurospheres were incubated at RT for 15 min in Accumax (Stem Cell Technologies) prior to quenching in neurobasal medium, centrifugation at $100 \times g$, and plating at $3 \times 10^4 - 1 \times 10^5$ densities. Cells were cultured 3–5 DIV.

**Live and fixed immunofluorescent cell imaging.** For live phase contrast or differential interference contrast (DIC) microscopy, images were acquired using either a ×10/0.3, ×20/0.5, or ×40/1.3 numerical aperture (NA) objective lens on a Nikon TE2000 inverted microscope with a Coolsnap HQ2 camera (Roper Scientific) with no binning (pixel size = 155 nm). Fixed immunostained images were acquired using either a ×60/1.45 NA objective lens at ×2–2.5 zoom on an Olympus Fluoview 500 laser-scanning confocal system mounted on an AX-70 microscope or using a ×63/1.4 NA objective lens at ×2–2.5 zoom on a Zeiss LSM800 Airy-disc scanning confocal system mounted on an Axio Observer Inverted Microscope. Live images of fluorescent neurons were acquired using the Nikon TE2000 or Zeiss microscopes.

**Immunofluorescence staining and outgrowth analyses.** Immunofluorescence (IF) staining was performed as in Santiago-Medina et al. 2011 and 2013[75,76]. hFB or hMN cultures were fixed in 4% paraformaldehyde in Krebs-sucrose fixative (4% PKS)[77]. Cells were then permeabilized in 0.1% Triton X-100 for 15 min, followed by incubation in a blocking solution of 1% fish gelatin and sodium azide in calcium-and-magnesium-free PBS for 1 h at room temperature. Primary antibodies were incubated at 4 °C overnight, and Alexa-Fluor-conjugated secondary antibodies were used at 1:250 dilution for 1 h at room temperature (Antibodies and dilutions are available in Supplemental Table 3). Measurements of fluorescence intensity of proteins of interest were made within growth cones by creating an image mask from a thresholded F-actin (AlexaFluor (AF)-phalloidin, Thermo-Fisher) channel and measuring the average pixel intensity of immunolabeling within the selected area. Measurements of fixed neurite lengths were made by tracing from the end of the neurite as determined by acetylated-tubulin (Sigma-Aldrich) labeling to the edge of the neurosphere using the Simple Neurite Tracer plug-in[78] and Fiji/ImageJ[79–81]. Image stacks of live neurons were stabilized as needed prior to analysis using the ImageStabilizer plug-in[82].

Live DIC outgrowth analysis was performed by taking consecutive measurements at the furthest edge of the growth cone lamellipodia opposite the central domain; retraction distances were capped at three times growth cone length of 30 microns to account for skewing of single very large retractions. The collapse was quantified via the disassembly of the lamellipodial veil and ≤2 filopodia of <10 microns in length.

All other data analysis was performed on raw images; the Unsharp Mask filter (ImageJ) was utilized in some instances for display purposes only. Gaussian blur background subtraction was similarly utilized in DIC images as needed for background shading correction for display purposes.

**Western blot.** Immunoblotting performed as in Basu et al.[63] using lysates collected as in Kerstein et al.[83]. Cell lysates were collected as neurosphere suspensions or as cultured cells incubated in RIPA buffer (ThermoFisher) supplemented with PhoSTOP (Roche) and Compleat protease inhibitor (Roche) followed by protein concentration evaluation (BCA Assay, Pierce/ThermoFisher) via a NanoDrop 2000 spectrophotometer. Approximately 20 μg of protein were run for each lane on a Bio-Rad 4–15% SDS-PAGE gel (Bio-Rad). Primary antibodies were used as follows: TSC2, phospho-S6, Actin (1:1000; Sigma-Aldrich), and RhoA (1:1000, Cell Signaling Technologies). Horseradish peroxidase-conjugated secondary antibodies (Jackson Immunoresearch) were used at 1:10000 and blots visualized via Pierce ECL and Pierce Femto ECL kits (ThermoFisher) (Additional information on antibodies and dilutions are available in Supplemental Table 3).

**Puromycin translation assay.** The assay was performed as in Schmidt, Clavarino, Ceppi, and Pierre, 2009[32]. Cultures were incubated in 10 μg/mL puromycin (ThermoFisher) for 10 min increments concurrent with or following treatment with Slit-2 (200 ng/mL), netrin-1 (100 ng/mL), or ephrin-A1 (200 ng–2 μg/mL) (R&D), and pharmacological agonists/antagonists as noted in the results, including rapamycin (Calbiochem) and anisomycin (MP Biomedicals). Puromycin-containing media was rinsed 2× with NB media before fixation. Growth cones were fixed and stained with anti-puromycin primary antibody (1:1000, PMY-2A4 was deposited to the DSHB by Yewdell, Jonathan (DSHB Hybridoma Product PMY-2A4)) and AF488 secondary (ThermoFisher) with AF-conjugated phalloidin (ThermoFisher) counterstain and analyzed as described above. All growth cones were normalized to $TSC2^{+/+}$ puromycin-only control.

**G-LISA Assay for RHOA activity.** Neurospheres were stimulated with ephrin-A1 for 15 min, then rinsed immediately in ice-cold phosphate-buffered saline (PBS, Sigma) prior to lysate collection. Cells were stored in G-LISA lysis buffer with manufacturer's protease and phosphatase inhibitors (Cytoskeleton). Protein concentrations were determined using Precision Red Advanced Protein Assay Reagent with absorbance at 600 nm measured using a NanoDrop 2000 spectrophotometer. Lysates were incubated in wells for 1 h at 4 °C with shaking, then rinsed and sequentially incubated with primary antibodies to RHOA followed by HRP secondary antibodies (Cytoskeleton), and the colorimetric reaction was performed for 15 min at 37 °C, which was measured using a Molecular Devices SpectraMax 250.

**iPSC nucleofection protocol.** iPSC neurons were transfected as in the Amaxa Primary Mammalian Neuron protocol (https://bioscience.lonza.com/lonza_bs/US/en/download/product/asset/21464) using whole or dissociated neurospheres, as described above. Cells were immersed in transfection solution with 1–5 μg of plasmid DNA and transfected using an Amaxa Nucleofector II, program settings O-O5 for dissociated cultures, and B-016 for neurospheres (kit VPI-1003, Amaxa). We utilized both Amaxa's commercial solution and a lab-prepared solution, noting no significant differences in performance between them in our hands. Cells were allowed to recover in suspension for 24 h before culturing on PDL-LN coated coverslips or on stripes (below).

**Lentiviral transduction assay.** iPSC neurons were transduced for 48 h at differentiation day 24–30 with RHOA-WT-RFP (LVP260) or control RFP (CMV-Null-RB) under a CMV promoter, acquired from GenTarget. Following 72 h recovery time, neurospheres were plated, individual neurites were visually screened for the presence of RFP, and growth cones were assayed for outgrowth and cue response.

**FRET assay.** To measure basal RHOA activity in growth cones, acceptor photobleaching was performed on neurons expressing a RHOA FRET sensor[48]. CFP (donor, excited with 440 nm diode laser) and YFP (acceptor, excited with 514 nm argon laser) images of live hFB neurons expressing a RHOA FRET sensor were collected before bleaching. Next, YFP fluorescence was locally bleached by repetitive fast scanning 520 nm light at 100% power for 5 s. Immediately after YFP bleaching, post-bleaching CFP and YFP images were collected with the same detector settings as used for original images. CFP emission typically increases when the acceptor (YFP) is bleached and FRET efficiency is recorded as the change in CFP/YFP.

**Stripe guidance assay.** Stripe molds produced by Dr. Martin Bastmeyer at Karlsruhe Institute of Technology were purchased or provided by Paul Letourneau. Acid-washed coverslips were first coated in PDL (50 μg/mL) for 45 min and rinsed extensively with ddH2O before drying. Molds were affixed and 50 μL of guidance cue plus LN (25 μg/mL) and dextran conjugated to tetramethylrhodamine (TMR) or AF647 (500 μg/mL; ThermoFisher) diluted in PBS was vacuum fed through the mold and allowed to incubate for one hour at RT, with one solution exchange. After extensive rinsing with PBS, molds were removed and the patterns were rinsed further with PBS. The cue-patterned coverslips were next flooded with LN (25 μg/mL) for one hour at RT followed by final PBS rinses. Cues utilized in stripe assay included ephrin-A1 (10 μg/mL; R&D Systems), ephrin-A5 (10 μg/mL; R&D), netrin1 (100 ng/mL; R&D), and hSlit2 (1 μg/mL; R&D). Cue binding was independently confirmed with antibodies to His (Netrin and Slit2) or Fc (ephrin-A5 and ephrin-A1) and visualization with appropriate AF-conjugated secondary. Ephrin-A5 stripe preparation required pre-clustering with anti-Fc antibody incubated under mild agitation at RT for 30 min before application. Unclustered ephrin-A5 stripes were utilized as a control.

Neurons on patterned cues were labeled AF-Phalloidin and neurite orientation and length within quadrants were quantified using a Sholl analysis plug-in[27]. In addition, image thresholding of the F-actin channel was used to measure the area of neurites on vs. off stripes. Sampled image analysis was confined to areas corresponding to the parallel quadrants and a minimum of 100 microns away from the neurosphere. Data were normalized to the mean neurite area occupying LN stripes.

**Reporting summary.** Further information on research design is available in the Nature Research Reporting Summary linked to this article.

## Data availability
A reporting summary for this article is available as a Supplementary Information file. The data supporting the findings of this study are available within the article, its Supplementary Figures, and source data file. Patient-derived and engineered TSC2 lines described in the paper will be made available at WiCell (https://www.wicell.org/). Source data are provided with this paper.

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

## Acknowledgements

We thank Erik Dent, Christopher Gomez, Kate Kalil, and members of the Gomez lab for comments on the manuscript. Thanks to the Dent, Huang, Kalil, and Roopra labs for reagents and advice. Also, this work would not have been possible if not for the inspiration and early guidance by Dr. Manuel R. Gomez, whose foundational work on tuberous sclerosis set the stage for these molecular studies. This work was supported, in part, by NIH Grant R21088477 and R01NS099405 (to T.M.G.). T.S.C. received support from grant T32 GM007215. A.J.M. and M.M.O. were supported in part by the University of Wisconsin-Madison Hilldale Undergraduate Research Fellowships.

## Author contributions

T.S.C. and T.M.G. designed the study; T.S.C., M.M.O., A.J.M., and S.K.R. performed the research and analyzed the data; J.G. and D.N.F. provided the patient samples. T.S.C. and T.M.G. wrote the paper.

## Competing interests

The authors declare no competing interests.
