## [Peer Review File · Nature Communications]

Reviewers' Comments:

Reviewer #1:

Remarks to the Author:

In Catlett et al.'s manuscript entitled "Abnormal axon guidance of human tuberous sclerosis neurons is due to mTOR-independent defects in RhoA signaling", the authors examine axon growth/guidance properties of TSC2 mutant patient neurons, which has not been examined before. In fact, much of the literature of ASD patient-derived iPSC-neuron field has focused on dendritic and synaptic mechanisms, and less so on axon guidance or growth mechanisms. The authors show an interesting mTOR-independent function of TSC2 in axon guidance through regulation of RhoA activity. This is somewhat unexpected since TSC2 is primarily thought to regulate translation, which also occurs locally in axons. However, the authors determine that heterozygous deletion of TSC2, mimicking most of the patient mutations, does not only impact translation but also mediates guidance/growth mechanisms through RhoA, a known effector of the actin cytoskeleton. The experiments include wildtype, heterozygous and homozygous TSC knockout human neurospheres, providing good evidence for their conclusion. This data is also interesting because it brings into question TSC2 mechanisms that are independent of translation, which might be contributing to other cellular events. Further, a number of control experiments are done, in particular for the guidance cue experiments. The conclusions from this manuscript will challenge others in the field to examine axon guidance/growth in other ASD iPSC lines since axon deficits are well known to exist in many other ASD models. However, there are outstanding issues that require further clarification and/or new experiments given the current iPSC field.

- 1) One major issue is the use of a single patient line, although it is CRISPR-corrected making it isogenic. But how do we know their conclusions apply to other TSC2 patient lines? Most studies now require at least 2 independent lines to verify data. Without some additional data in a second line (genetically independent), it's difficult to make strong conclusions. There is a lot of line-to-line variability when performing iPSC experiments.
- 2) The neurosphere assays do not have quantification or any indication of how big the neurospheres (size) are when being used for the extension rate on axon guidance cues. It would seem that bigger neurospheres could have faster extension rates because more neuron/axons are present. The authors should show that the size of the neurosphere is not a confounding factor for their results, or show that similar sized neurospheres were in fact used in their assays.
- 3) How do the authors know that all of the neurites growing out of the neurospheres are indeed axons? Could some of their measurements include dendrites? Simple markers could be used to address this.
- 4) The authors do not provide details on how many neurites/neurospheres were used in each experiment. If each data point refers to a neurite, then neurites from the same neurosphere should instead be considered technical replicates and not its own biological replicate for each experiment.
- 5) The authors show imaging of FOXG1 in supplemental figure 1, however they do not provide an explanation of why they did this staining and what they were trying to convey from the image.
- 6) Can the axon growth/guidance defects be rescued by pharmacologically or genetically manipulating RhoA?
- 7) It is unclear why both cortical and motor neurons were used, and if there are different conclusions from using both approaches? Are growth cones similar between the two populations, and their responses to guidance cues?
- 8) The authors mention that 6 clones were generated from this patient, but they don't mention how many clones they used for the experiments. Since only one patient was used, it would be important to use more than one clone.
- 9) The authors do not show any type of staining or western blot to show that the human motor neurospheres were indeed made of motor neurons. They could possibly stain for Olig2 or Islet1/2.
- 10) Line 150 is unclear, I think the subtitle should read "Guidance of cortical axons by patterned inhibitory cues is defective in TSC2+/- neurons"

11) In Figure 4, the authors should label D and E as pre and post ephrin treatment

Reviewer #2:

Remarks to the Author:

Gomez and colleagues set out to investigate the molecular basis for axon misguidance in human iPSC-derived neurons. For this purpose, they differentiate skin fibroblasts from a TSC patient into forebrain (hFB) and motor neurons (MNs). In addition to the patient-derived clones (Tsc2+/-), they generate a CRISPR-corrected isogenic control TSC2+/+ iPSC lines and introduce the same mutation in the unaffected allele of the heterozygous patient line to create a null TSC2-/- iPSC line. The authors show that basal mTOR activity and mTOR-dependent protein synthesis are unchanged in the heterozygous neurons but are both increased in the TSC2 null hFBs neurons. In addition, while the human neurons derived from unaffected controls and disease-corrected iPSCs respond to canonical guidance cues, the Tsc2+/- neurons have reduced responsiveness to repulsive cues and defective axon guidance. They also find enhanced basal axon extension in the Tsc2+/- neurons and show that mTORC1 and mTORC2 inhibitors do not rescue these defects. To identify the mTOR-independent mechanism responsible for altered axonal guidance, they investigate RhoA activity and find that both basal and cue-activated RhoA activity are reduced in the heterozygous neurons, suggesting that cue-activated RhoA is necessary for proper axon growth and guidance. Interestingly, they find reduced phosphorylation of the RhoA downstream target the myosin light chain MLC and further show that pharmacological inhibition of RhoA in control neurons resembles the phenotypes of the TSC2+/- axons while re-activation of RhoA in the TSC2+/- rescued both the increased axonal extension and cue-responsiveness. They conclude that rapid TSC2-dependent protein synthesis changes are not required for outgrowth and cue responses, but direct RhoA signaling downstream of TSC1/TSC2 is necessary for proper axonal development.

Overall, the manuscript is very interesting and provides novel insights into the regulation of axon guidance in iPSC-derived human neuronal cultures. The experimental design is appropriate, and the experiments are well executed. There are some issues that can be addressed to increase the impact of the findings.

Specific comments:

The authors state that six iPSC clones of each genotype were generated; however, they do not specify how many clones were used for the neuronal differentiation. This information together with the technical replica and passage number of the clones used should be included for each experiment.

Page 6: The authors state: "heterozygous loss of TSC2 does not appear to impact basal mTOR activity within growth cones nor within neurospheres, which is consistent with other studies performed in rodent and iPS neuronal models^{23–25}." This is not a completely accurate interpretation of the literature. There are several reports of both mouse and iPSC-derived human neurons with TSC1 or TSC2 heterozygous genotype that show increase mTORC1 activity in the literature (e.g. recent Martin et al., Mol Autism 2020; Zucco et al., MCN 2018; Winden et al., J Neurosci 2019; Ehninger et al., Nature Med 2008, etc...). Thus, the authors should qualify their conclusion that "heterozygous loss of TSC2 does not appear to impact basal mTOR activity within growth cones nor within neurospheres" by adding "under these culture conditions."

Fig. 5A-B: Here the authors show the protein synthesis rates in the growth cones in response to different cues. The quantification shows a significant increase in the growth cones of the TSC2-/-, but the IF does not seem to reflect the quantification. How were the data normalized?

Fig. 5C: The authors should show a representative immunofluorescent image of the effect of EphrinA1 in the TSC2-/- neurons.

Fig. 7: It is shown that RhoA activity is diminished in both the mutant hFB neurons and neurospheres. Did the authors assess expression level of RhoA?

Fig. 8: The authors argue that activation of MLC with CalyA sensitizes TSC2^{+/-} neurons to ephrin-A1 mediated collapse. They need to show the data that CalyA by itself does not cause grown cone collapse.

Fig 8D-E: This experiment shows that control neurons with ROCK inhibitor treatment showed no significant substratum preference on the stripe assay, similar to TSC2^{+/-} neurons. This by itself does not prove that RhoA activity is necessary for EphrinA-mediated axon guidance. A more informative experiment would be to treat the neurons on the stripe assay with CalyA and see if this leads to TSC2^{+/-} axons avoiding the ephrin stripes.

Minor:

Authors should add information of karyotyping of the iPSCs lines in the materials and methods section and supplemental data.

The quality control data of iPSC-colonies immunocytochemistry were not included in the paper, although they were mentioned in the material and methods section. Authors should add the images of immunostained iPSC colonies with OCT4 and NANOG into supplemental data. Also, additional markers of pluripotency like Tra1-60 or Tra1-81 should be used to verify the pluripotency of the cells after gene-editing.

Fig 4d: In the result section (page 13), it is erroneously stated that this figure shows lack of responsiveness in response to Slit2; however, the figure to refer is 4F.

Mustafa Sahin

Reviewers' comments:

Reviewer #1 (Remarks to the Author):

In Catlett et al.'s manuscript entitled "Abnormal axon guidance of human tuberous sclerosis neurons is due to mTOR-independent defects in RhoA signaling", the authors examine axon growth/guidance properties of TSC2 mutant patient neurons, which has not been examined before. In fact, much of the literature of ASD patient-derived iPSC-neuron field has focused on dendritic and synaptic mechanisms, and less so on axon guidance or growth mechanisms. The authors show an interesting mTOR-independent function of TSC2 in axon guidance through regulation of RhoA activity. This is somewhat unexpected since TSC2 is primarily thought to regulate translation, which also occurs locally in axons. However, the authors determine that heterozygous deletion of TSC2, mimicking most of the patient mutations, does not only impact translation but also mediates guidance/growth mechanisms through RhoA, a known effector of the actin cytoskeleton. The experiments include wildtype, heterozygous and homozygous TSC knockout human neurospheres, providing good evidence for their conclusion. This data is also interesting because it brings into question TSC2 mechanisms that are independent of translation, which might be contributing to other cellular events. Further, a number of control experiments are done, in particular for the guidance cue experiments. The conclusions from this manuscript will challenge others in the field to examine axon guidance/growth in other ASD iPSC lines since axon deficits are well known to exist in many other ASD models. However, there are outstanding issues that require further clarification and/or new experiments given the current iPSC field.

1) One major issue is the use of a single patient line, although it is CRISPR-corrected making it isogenic. But how do we know their conclusions apply to other TSC2 patient lines? Most studies now require at least 2 independent lines to verify data. Without some additional data in a second line (genetically independent), its difficult to make strong conclusions. There is a lot of line-to-line variability when performing iPSC experiments.

To address this concern, CRISPR/Cas9 was used to edit IMR90-4 control iPSC's to create an isogenic premature stop codon in one allele of TSC2 at the same site of our patient line. The advantage here is that this control line was already used throughout our paper. Key experiments were repeated in these lines to first ensure that TSC2 was present in haploinsufficient quantities in growth cones, as well as to replicate outgrowth, cue response, and guidance phenotypes (Figure 3, Supplemental Figures 1,2, 4). By mutating this well-established iPSC line, we added important support to our conclusions that TSC2 heterozygosity via nonsense mutation in human stem cells is sufficient to cause defects in neurite behavior.

2) The neurosphere assays do not have quantification or any indication of how big the neurospheres (size) are when being used for the extension rate an axon guidance cues. It would seem that bigger neurospheres could have faster extension rates because more neuron/axons are present. The authors should show that the size of the neurosphere is not a confounding factor for their results, or show that similar sized neurospheres were in fact used in their assays.

Reviewer #1 raises a good point. Within our assays, we saw no difference in extension rates from neurospheres between 200-1000 μ m in diameter and did not use neurospheres outside these boundaries (Supplemental Figures 4A). There was also no significant difference in TSC2+/+ and TSC2+/- neurosphere sizes (was this shown anywhere in paper?). Furthermore, very large neurospheres within all groups often represented aggregations of smaller neurospheres and did not adhere well to the coverslips and exhibited poor neurite extension. To indicate this, we have added clarifying language and additional quantification of neurite length/neurosphere size in each genotype to Supplemental Figure 4A, showing that neurite lengths were not correlated at the neurosphere sizes utilized in this study.

3) How do the authors know that all of the neurites growing out of the neurospheres are indeed axons? Could some of their measurements include dendrites? Simple markers could be used to address this.

In this study, we specifically excluded neurites that were shorter than 100 μ m to eliminate the confluence of axons/dendrites. All long processes with cell bodies embedded within the neurospheres stained positive for tau, and while early neurites weakly stain for MAP2, strong MAP2 staining associated with dendritic process appeared outside the neurosphere at approximately two weeks in culture and were restricted to shorter processes. We have included examples of these differences in Supplemental Figure 1E.

4) The authors do not provide details on how many neurites/neurospheres were used in each experiment. If each data point refers to a neurite, then neurites from the same neurosphere should instead be considered technical replicates and not its own biological replicate for each experiment.

To address this concern, we have added more complete information as to the number of neurospheres utilized per experimental group within each experiment. This supporting data is extensive for some of the quantifications shown in the main figures and has been aggregated in its entirety in Supplemental Table 1. For each experimental group, four or more neurospheres were used unless otherwise specified.

5) The authors show imaging of FOXG1 in supplemental figure 1, however they do not provide an explanation of why they did this staining and what they were trying to convey from the image.

To correct this oversight, we have added clarifying language within Supplemental Figure 1 specifying the use of FOXG1 as a forebrain marker, in addition to language within the Methods section.

6) Can the axon growth/guidance defects be rescued by pharmacologically or genetically manipulating RhoA?

While pharmacological agonists of RhoA itself are not available, we were able to address this question in multiple ways. Pharmacological manipulation of myosin phosphatase downstream of RhoA partially rescued outgrowth and collapse phenotypes within TSC2+/- neurites (Figure 8B,C). To more directly examine the role of RhoA, we transduced TSC2+/- neurites with a lentiviral RhoA-RFP construct in addition to an RFP-only control. We found that RhoA over-expression was sufficient to decrease

outgrowth in TSC2+/- neurites, and to sensitize these growth cones to ephrin-A1 (Figure 8B,C). RhoA over-expression was also confirmed to increase myosin activity in neurons (Supplemental Figure 8E,F).

7) It is unclear why both cortical and motor neurons were used, and if there are different conclusions from using both approaches? Are growth cones similar between the two populations, and their responses to guidance cues?

We performed preliminary experiments in MNs to explore the possibility that the phenotypes we observed may generalize to other classes of neurons, which also allowed us to test additional guidance cues. While limiting space does not allow us to explore this in depth, the selective cue resistance by MN's support the robustness of TSC2 pathway-mediated axon guidance and suggests that this may be a fruitful area for future investigation in the community.

8) The authors mention that 6 clones were generated from this patient, but they don't mention how many clones they used for the experiments. Since only one patient was used, it would be important to use more than one clone.

With respect for the variability of iPS cell lines, we performed all key experiments using two clones per TSC patient cell line (TSC2+/, TSC2+/-, and TSC2-/-), and have updated the manuscript to clarify this, and as mentioned above, also included additional data in Supplemental Table 1. Additional clones from acquired WA09 and IMR90 cells were not available. Importantly, with the inclusion of new experimental data from a second heterozygous TSC2 line, we are confident that our reported results are generalizable to TSC2 heterozygous neurons.

9) The authors do not show any type of staining or western blot to show that the human motor neurospheres were indeed made of motor neurons. They could possibly stain for Olig2 or Islet1/2.

We have added our quantification of the MN-specific marker Hb9 within TSC2+/+ and TSC2+/- neurons to Supplementary Figure 3.

10) Line 150 is unclear, I think the subtitle should read "Guidance of cortical axons by patterned inhibitory cues is defective in TSC2+/- neurons"

We agree and have adjusted the language accordingly.

11) In Figure 4, the authors should label D and E as pre and post ephrin treatment

We have corrected this oversight in the revised manuscript.

Reviewer #2 (Remarks to the Author):

Gomez and colleagues set out to investigate the molecular basis for axon misguidance in human iPSC-derived neurons. For this purpose, they differentiate skin fibroblasts from a TSC patient into forebrain (hFB) and motor neurons (MNs). In addition to the patient-derived clones (Tsc2+/-), they generate a CRISPR-corrected isogenic control TSC2+/+ iPSC lines and introduce the same mutation in the unaffected allele of the heterozygous patient line to create a null TSC2-/- iPSC line. The authors show that basal mTOR activity and mTOR-dependent protein synthesis are unchanged in the heterozygous neurons but are both increased in the TSC2 null hFBs neurons. In addition, while the human neurons derived from unaffected controls and disease-corrected iPSCs respond to canonical guidance cues, the Tsc2+/- neurons have reduced responsiveness to repulsive cues and defective axon guidance. They also find enhanced basal axon extension in the Tsc2+/- neurons and show that mTORC1 and mTORC2 inhibitors do not rescue these defects. To identify the mTOR-independent mechanism responsible for altered axonal guidance, they investigate RhoA activity and find that both basal and cue-activated RhoA activity are reduced in the heterozygous neurons, suggesting that cue-activated RhoA is necessary for proper axon growth and guidance. Interestingly, they find reduced phosphorylation of the RhoA downstream target the myosin light chain MLC and further show that pharmacological inhibition of RhoA in control neurons resembles the phenotypes of the TSC2+/- axons while re-activation of RhoA in the TSC2+/- rescued both the increased axonal extension and cue-responsiveness. They conclude that rapid TSC2-dependent protein synthesis changes are not required for outgrowth and cue responses, but direct RhoA signaling downstream of TSC1/TSC2 is necessary for proper axonal development.

Overall, the manuscript is very interesting and provides novel insights into the regulation of axon guidance in iPSC-derived human neuronal cultures. The experimental design is appropriate, and the experiments are well executed. There are some issues that can be addressed to increase the impact of the findings.

Specific comments:

The authors state that six iPSC clones of each genotype were generated; however, they do not specify how many clones were used for the neuronal differentiation. This information together with the technical replica and passage number of the clones used should be included for each experiment.

We have expanded the data where appropriate to reflect that two clones from each patient-derived genotype were used for all neuronal differentiations and extended data for all experimental replicates were added to Supplemental Table 1. Also, we did not observe significant variation in phenotypes across passages up to p60, but did not include experiments on cultures beyond p50 within this manuscript. We also have added outgrowth data from all tested lines at early and late passages to Supplementary Figure 4B, showing no significant intra-line variation.

Page 6: The authors state: “heterozygous loss of TSC2 does not appear to impact basal mTOR activity within growth cones nor within neurospheres, which is consistent with other studies performed in rodent and iPS neuronal models^{23–25}.” This is not a completely accurate interpretation of the literature. There are several reports of both mouse and iPSC-derived human neurons with TSC1 or TSC2 heterozygous genotype that show increase mTORC1 activity in the literature (e.g. recent Martin et al.,

Mol Autism 2020; Zucco et al., MCN 2018; Winden et al., J Neurosci 2019; Ehninger et al., Nature Med 2008, etc...). Thus, the authors should qualify their conclusion that "heterozygous loss of TSC2 does not appear to impact basal mTOR activity within growth cones nor within neurospheres" by adding "under these culture conditions."

We have corrected this oversight, clarifying as suggested and adding additional citations within the discussion.

Fig. 5A-B: Here the authors show the protein synthesis rates in the growth cones in response to different cues. The quantification shows a significant increase in the growth cones of the TSC2^{-/-}, but the IF does not seem to reflect the quantification. How were the data normalized?

All treatment groups were normalized to puromycin-conjugated fluorescence intensity within TSC2^{+/+} growth cones under basal conditions, averaged to growth cone size. We also performed initial pilot experiments normalizing to the total protein marker SE647 to confirm that total protein levels between growth cones was not a significant confounding factor in measuring differences in local protein synthesis between growth cones. Despite size variation within individual growth cones, there were no significant differences in growth cone size between groups. We have also updated the language within the results and methods to more accurately reflect these parameters. The original TSC2^{-/-} example was 2x in intensity relative to TSC2^{+/+} and TSC2^{+/-}; we have updated this image to more accurately reflect the quantification.

Fig. 5C: The authors should show a representative immunofluorescent image of the effect of EphrinA1 in the TSC2^{-/-} neurons.

We have reorganized Figure 5 to reflect this.

Fig. 7: It is shown that RhoA activity is diminished in both the mutant hFB neurons and neurospheres. Did the authors assess expression level of RhoA?

To address this question, we performed WB analysis of RhoA within TSC2^{+/+} and TSC2^{+/-} neurospheres and found no significant differences in RhoA expression between these groups. This data is now shown in Supplementary Figure 8C,D.

Fig. 8: The authors argue that activation of MLC with CalyA sensitizes TSC2^{+/-} neurons to ephrin-A1 mediated collapse. They need to show the data that CalyA by itself does not cause growth cone collapse.

This was a key issue within our experiments, as we found that higher doses of CalyculinA does rapidly collapse TSC2^{+/-} and TSC2^{+/+} growth cones. By titrating the dose of Calyculin to pM levels, we were able to sensitize growth cones to ephrin-A1 treatment without collapsing them. We have updated Supplementary Figure 8A,B to reflect this additional CalyculinA-only collapse data.

Fig 8D-E: This experiment shows that control neurons with ROCK inhibitor treatment showed no significant substratum preference on the stripe assay, similar to TSC2+/- neurons. This by itself does not prove that RhoA activity is necessary for EphrinA-mediated axon guidance. A more informative experiment would be to treat the neurons on the stripe assay with CalyA and see if this leads to TSC2+/- axons avoiding the ephrin stripes.

We agree that CalyculinA treatment would have been the more informative experiment. However, in our hands chronic exposure to low concentrations of CalyculinA resulted in unhealthy cultures. While partial rescue was sometimes observed, the data were not included because we could not confirm unintentional effects from the possible cytotoxicity. We have included this information in the revised manuscript. As an alternative strategy, we transduced neurons with RhoA via lentivirus as outlined above. While effective in outgrowth and acute cue response assays, the low efficiency of RhoA-WT overexpression in neurons prevented observation of cell autonomous guidance rescue.

Minor:

Authors should add information of karyotyping of the iPSCs lines in the materials and methods section and supplemental data.

Karyotyping was completed on all cell lines by WiCell, Madison WI. We have included additional karyotyping information to the materials and methods section.

The quality control data of iPSC-colonies immunocytochemistry were not included in the paper, although they were mentioned in the material and methods section. Authors should add the images of immunostained iPSC colonies with OCT4 and NANOG into supplemental data. Also, additional markers of pluripotency like Tra1-60 or Tra1-81 should be used to verify the pluripotency of the cells after gene-editing.

We have included representative images of OCT4, NANOG, and Tra1-60 of each genotype to Supplementary Figure 1.

Fig 4d: In the result section (page 13), it is erroneously stated that this figure shows lack of responsiveness in response to Slit2; however, the figure to refer is 4F.

We have corrected this oversight in the revised manuscript.

Mustafa Sahin

Reviewers' Comments:

Reviewer #1:

Remarks to the Author:

The authors have done a very good job of addressing my concerns, most notably generating a second line and perform axonal experiments to corroborate their results. It seems they also addressed Rev 2's comments well in my opinion. This is one of the first studies examining the role of TSC2 in human axon growth and will be of importance to the field. I strongly support publication.

Reviewer #2:

Remarks to the Author:

The authors have addressed the issues that I have raised and have provided a thoughtful response to my critique.

Mustafa Sahin